



# Inferring iron oxides species content in atmospheric mineral dust from DSCOVR EPIC observations

Sujung Go[1,2], Alexei Lyapustin[2], Gregory L. Schuster[3], Myungje Choi[1,2], Paul Ginoux[4], Mian Chin[2], Olga Kalashnikova[5], Oleg Dubovik[6], Jhoon Kim[7,8], Arlindo da Silva[2], Brent Holben[2] and Jeffrey S. Reid[9]

[1] University of Maryland Baltimore County, Baltimore, MD, USA
[2] NASA Goddard Space Flight Center, Greenbelt, MD, USA
[3] NASA Langley Research Center, Hampton, VA, USA
[4] Geophysical Fluid Dynamics Laboratory, Princeton, NJ, United States
[5] Jet Propulsion Laboratory, California Institute of Technology, Pasadena, CA, USA
[6] Laboratoire d'Optique Atmosphérique, Université de Lille-1, CNRS, Villeneuve d'Ascq, France
[7] Yonsei University, Seoul, Korea
[8] Particulate Matter Research Institute, Samsung Advanced Institute of Technology, Korea
[9] US Naval Research Laboratory, Monterey, CA, USA

*Correspondence to*: Sujung Go (sujung.go@nasa.gov)

**Abstract.** The iron-oxide content of dust in the atmosphere and most notably its apportionment between hematite ($\alpha$-Fe$_2$O$_3$) and goethite ($\alpha$-FeOOH) are key determinants in quantifying dust's light absorption, its top of atmosphere UV radiances used for dust monitoring, and ultimately shortwave dust direct radiative effects (DRE). Hematite and goethite column mass concentrations and iron-oxide mass fractions of total dust mass concentration were retrieved from the Deep Space Climate Observatory (DSCOVR) Earth Polychromatic Imaging Camera (EPIC) measurements in the ultraviolet–visible (UV–Vis) channels. The retrievals were performed for dust-identified aerosol plumes using aerosol optical depth (AOD) and spectral imaginary refractive index provided by the Multi-Angle Implementation of Atmospheric Correction (MAIAC) algorithm over six continental regions (North America, North Africa, West Asia, Central Asia, East Asia, and Australia). The dust particles are represented as an internal mixture of non-absorbing host and absorbing hematite and goethite. We use the Maxwell–Garnett effective medium approximation with carefully selected complex refractive indices of hematite and goethite that produce mass fractions of iron oxides species consistent with *in situ* values found in the literature to derive the hematite and goethite volumetric/mass concentrations from MAIAC EPIC products. We compared the retrieved hematite and goethite concentrations with *in situ* dust aerosol mineralogical content measurements, as well as with published data. Our data display variations within the published range of hematite, goethite, and iron-oxide mass fractions for pure mineral dust cases. A specific analysis is presented for 15 sites over the main dust source regions. Sites in the central Sahara, Sahel, and Middle East exhibit a greater temporal variability of iron oxides relative to other sites. Niger site (13.52°N, 2.63°E) is dominated by goethite over Harmattan season with median of ~2 weight percentage (wt.%) of iron-oxide. Saudi Arabia site (27.49°N, 41.98°E) over Middle East also exhibited surge of goethite content with the beginning of Shamal season. The Sahel dust is richer in iron-oxide than Saharan and northern China dust except in Summer. The Bodélé Depression area shows a distinctively lower iron-oxide concentration





(~1 wt.%) throughout the year. Finally, we show that EPIC data allow to constrain the hematite refractive index. Specifically,
we select 5 out of 13 different number of hematite refractive indices widely variable in published laboratory studies by
constraining the iron-oxide mass ratio to the known measured values. Provided climatology of hematite and goethite mass
fractions across main dust regions of the Earth will be useful for dust shortwave DRE studies and climate modeling.

# 1 Introduction

Aeolian dust, suspended in the troposphere at a rate of 1–4 Pg yr$^{-1}$, persists for 1–7 days or longer depending on
particle size (Boucher et al., 2013). Such dust emissions are caused by saltation in desert regions, seasonal river discharges,
and anthropogenic land use (e.g., overgrazing; Ginoux et al., 2012). Airborne dust contributes to the direct radiative effect
(DRE) by absorbing or scattering solar and terrestrial radiation in the shortwave (SW, 0.185–4.0 μm) and longwave (LW,
3.33–1,000 μm) spectral regions, respectively (Di Biagio et al., 2020). However, these dust absorption/scattering properties
may change substantially during transport with dust chemical composition and size distribution varying over different source
regions and with time. Accurate information on the space-time variability of dust spectral absorbing/scattering properties is
therefore crucial in estimating direct radiative forcing (Samset et al., 2018).

The dust DRE at the top of the atmosphere (TOA) is unresolved in both sign and magnitude. For example, Kok et al.
(2017) suggested that the net (SW + LW) dust DRE is cooling at −0.20 W m$^{-2}$ with an uncertainty range of −0.48 to +0.20 W
m$^{-2}$, based on the complex refractive index from Optical Properties of Aerosols and Clouds (OPAC) database (Hess et al.,
1998; Volz, 1973). Di Biagio et al. (2020) also suggested cooling at a lower rate of −0.03 W m$^{-2}$ with an uncertainty range of
−0.29 to +0.23 W m$^{-2}$, based on obtained complex refractive indices of different types of mineral dust (Di Biagio et al., 2017;
Balkanski et al., 2007). Li et al. (2021) determined a warming effect of +0.04 W m$^{-2}$ with an uncertainty range of −0.23 to
+0.35 W m$^{-2}$, based on CAM5 with a complex refractive index proposed by Scanza et al. (2015). Uncertainties in SW DRE
are generally greater than those of LW DRE. The DRE of "pure" mineral dust is determined mainly by its particle size
distribution, mineral composition, and particle shape (Sokolik and Toon, 1999; Knippertz and Stuut, 2014) as well as the
absolute dust concentration and the height and profile of the dust layer in the atmosphere. Previous studies have pointed out
that current climate models use a globally invariant spectral complex refractive index (and therefore spectral single scattering
albedo (SSA); Scanza et al., 2015; Samset et al., 2018; Di Biagio et al. 2019), which implicitly assumes the same dust
mineralogical composition on a global scale.

A few Earth-system models (ESM's, i.e., coupled climate models) have adopted regionally and temporally variable
spectral refractive index of dust by parameterization with common soil mineralogy components (Scanza et al. 2015; Perlwitz
et al. 2015a, b). The rationale for this is that dust aerosols are soil particles suspended in the atmosphere (Scanza et al., 2015).
Specifically, in CAM5 (Scanza et al., 2015; Liu et al., 2012), dust aerosol emission mineralogy is transformed from clay-sized
[diameters Dp = 0–2 μm] and silt-sized [Dp = 2–50 μm] particles in soil mineralogy to bimodal aeolian dust size distributions





(accumulation mode Dp = 0.1–1.0 µm; coarse mode Dp = 1–10 µm) by brittle fragmentation theory (Kok, 2011). The dust aerosol refractive index is then calculated using a volume-weighted mixing rule for all mineral components including water. Mineral components are internally mixed within each particle mode, and externally mixed between different particle modes (Liu et al., 2012, 2016). Finally, radiation is simulated by CAM5 for SW and LW spectral regions. The direct radiative forcing of dust is determined by the difference between the results of two calculations: radiative forcing with all aerosol species, and

radiative forcing with all aerosol species except mineral dust. In short, dust radiative forcing is highly dependent on mineral-specific dust absorption properties in the CAM5 ESM.

Li et al. (2021) recently quantified the importance of soil mineralogical content uncertainty on the dust DRE estimate using the soil atlases C1999 (Claquin et al., 1999) and J2014 (Journet et al., 2014), with the addition of goethite to C1999. They concluded that the iron-oxide fraction in dust represents 97% of the uncertainty in their estimated total dust DRE (−0.23

to +0.35 W m$^{-2}$) using CAM5 only, and 85% across multiple climate models. They also highlighted the importance of speciation of iron oxides into hematite and goethite to better estimate the SW dust DRE, without which the model would underestimate dust warming by 56% because the absorption magnitudes of hematite and goethite are up to an order of magnitude different at ultraviolet (UV) and visible (Vis) wavelengths (Fig. 1).

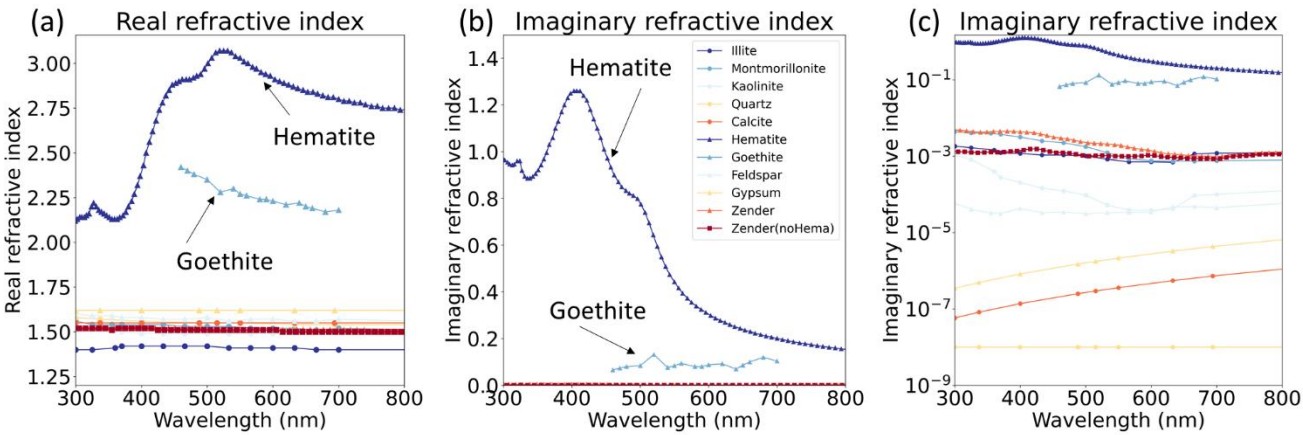

**Figure 1: Complex refractive index based on soil mineralogy at 300–800 nm (Scanza et al., 2015). Shown are the real refractive index (a) and the imaginary refractive index in linear (b) and log (c) scales. Goethite refractive index is from Bedidi and Cervelle (1993).**

Hematite (α-Fe$_2$O$_3$) and goethite (α-FeOOH), both in the Fe(III) oxidation state, are the major iron-oxide species (also referred together as "free iron" or "iron oxides") in mineral dust (Lafon et al., 2004), and exert major control on the absorption magnitude of pure dust for SW radiation (e.g., Sokolik and Toon,1999; Arimoto et al., 2002; Lafon et al., 2006; Formenti et

al., 2014a), as can be inferred from their complex imaginary refractive index (Fig. 1). The imaginary index magnitudes of both hematite and goethite (10$^{-1}$–10$^0$) are more than 100 times those of other soil mineral components (10$^{-8}$–10$^{-3}$) at wavelengths of <1 µm, which means that hematite and goethite dominate absorption while other minerals can be considered non-absorbing.


Hematite imaginary index (0.2 at 700 nm to 0.8 at 460 nm) is generally about three times higher than that of goethite (~0.1) (Bedidi and Cervelle, 1993) in the red–near infrared (NIR), the discrepancy further increasing towards blue–UV. At 350–450

nm, the imaginary index of hematite peaks at 1.0–1.2, whereas it changes little for goethite remaining at ~0.1. This significant difference in spectral absorption between hematite and goethite facilitates their separate retrievals.

Despite its high radiative impact on SW dust DRE, the relative proportion of iron oxides in the total dust mass is very small (up to 6.5 weight percentage-wt.%; Schuster et al., 2016). Iron oxides also control the color of soil (Torrent et al., 1983). A reddish color indicates hematite (hues of 5YR and 10R) and yellowish-brown colors indicate goethite (hues of 7.5YR

and 2.5Y) in soils, with the range of hues depending on the concentration, crystal size, and degree of cementation (Schwertmann, 1971; 1993). Hematite and goethite generally occur together in both soil and in the atmosphere, and they are internally mixed with other mineral particles, whereas most other dust minerals are externally mixed (Formenti et al., 2014a). Their formation in soil is influenced by climate change, with cooler and more humid conditions favoring goethite formation owing to changes in the organic-matter regime of the soil (Schwertmann, 1971).

The GRASP-components algorithm (Li et al., 2019) uses the Maxwell–Garnett effective medium approximation to provide aerosol speciation from the Polarization and Anisotropy of Reflectances for Atmospheric science coupled with Observations from a Lidar (PARASOL) developed for the POLarization and Directionality of the Earth's Reflectances (POLDER) program. Several future satellite missions such as Multi-Angle Imager for Aerosols (MAIA) (Diner et al., 2018) and Earth Surface Mineral Dust Source Investigation (EMIT) (Green et al., 2020) will be providing aerosol composition or

mineralogical information. The Earth Polychromatic Imaging Camera (EPIC) instrument has been operational since 2015, while POLDER/PARASOL was decommissioned in December 2013.

Several studies used regional or global AErosol RObotic NETwork (AERONET) inversion products (Arola et al., 2011; Koven and Fung, 2006; Schuster et al., 2005, 2016) or ground-based data (Li et al., 2013, 2015; Wang et al., 2013) to retrieve aerosol components.

This study retrieves hematite and goethite volume fractions and mass concentrations in atmospheric mineral dust from the EPIC UV–Vis (340, 388, 443, 680 nm) data, based upon the above physical characteristics of hematite and goethite. We began with Multi-Angle Implementation of Atmospheric Correction (MAIAC) retrieved spectral aerosol absorption information, i.e., the imaginary index at 680 nm ($k_0$) and spectral absorption exponent ($b$) (Lyapustin et al., 2021), currently optimized for "pure" smoke or dust.

The methodology is based on the Maxwell–Garnett effective-medium approximation and follows the work of Schuster et al. (2016). Assuming atmospheric aerosols are inhomogeneous particles with different complex dielectric functions, the Maxwell–Garnett approximation can be used to calculate average dielectric functions because of the interactions of electromagnetic waves between inhomogeneous particles (Bohren and Huffman, 1983). The theory considers randomly inhomogeneous mixtures comprised of two inclusions embedded in a homogeneous matrix (Bohren and Huffman, 1983). With





the host considered a homogeneous matrix with inclusions of hematite and goethite, we undertook component retrievals from
       dust-identified aerosol plumes with the MAIAC EPIC algorithm over six continental regions following Schuster et al. (2016)
       with some modifications. One issue with hematite is that its spectral refractive indices obtained in laboratory studies are varying
       widely (Schuster et al., 2016; Zhang et al., 2015). We examine how the volume fractions of hematite and goethite change with
       the hematite refractive index in Sections 2.3 and 4.

The remainder of this paper is organized as follows. Section 2 describes the input data and methodology; Section 3
       presents the results of selected case studies (Sec. 3.1), a comparison with *in situ* soil measurements (Sec. 3.2), and a summary
       of regional-seasonal climatology (Sec. 3.3); Section 4 presents elimination analysis of different hematite refractive index
       models consistent with EPIC UV–Vis measurements and *in situ* iron-oxide mass ratio. The results of this work are summarized
       in the concluding Section 5.

**2 Data and methodology**

       **2.1 v2 MAIAC EPIC Algorithm**

            The EPIC aboard the DSCOVR sattelite has been providing observations of the sunlit side of Earth from the first
       Lagrangian point (L1) since 2015. EPIC provides hourly measurements of the sunlit part of the Earth as it rotates from sunrise
       to sunset. Such a capability has not previously been available from any other spacecraft or Earth-observing platform. Following
the Moderate Resolution Imaging Spectroradiometer (MODIS) MAIAC algorithm (Lyapustin et al., 2018), the MAIAC EPIC
       algorithm over land provides cloud detection, retrieval of aerosol optical depth (AOD) with regionally specified background
       aerosol models, and atmospheric correction. Recently, we expanded the MAIAC EPIC algorithm to include simultaneous
       retrieval of AOD and spectral imaginary refractive index for the detected absorbing aerosols, e.g. biomass burning smoke and
       mineral dust (Lyapustin et al., 2021). Spectral absorption in MAIAC is represented by a power-law expression,

$k_\lambda = k_0 (\lambda/\lambda_0)^{-b}$ , where $\lambda_0 = 680$ nm                                                (1)

       where $k$ is an imaginary refractive index. The real refractive index and the size distribution for smoke and dust models are
       fixed. Parameters (AOD$_{443}$, $k_0$, $b$) are derived simultaneously by matching the EPIC TOA reflectance at 340, 388 and 443 nm.
       The surface reflectance at these wavelengths is parameterized using the EPIC red band (680 nm) and their spectral ratios to
       the red band obtained for each 10 km grid cell using the minimum reflectance method. The variety of retrieved combinations
($k_0$, $b$) characterize the magnitude of aerosol absorption and its spectral variability. MAIAC retrievals are reported for the
       effective aerosol heights of 1 km and 4 km, representing the typical boundary layer and free troposphere transported aerosol.
       As current MAIAC EPIC cannot discriminate between smoke and dust, it uses the dust model for known dust source regions
       (e.g., Sahara, Arabian Peninsula etc.), and the smoke model is applied elsewhere globally. The lack of the mixed smoke-dust





aerosol types is a current limitation of the algorithm. The MAIAC EPIC version 2 reports the combination of AOD, $k_0$ and $b$ along with calculated SSA at 443 nm. Our comparisons of SSA of MAIAC EPIC for 2018 for mineral dust with AERONET SSA showed an agreement with correlation coefficient R~0.62, RMSE ~0.021, and MBE ~0.006 over Sahara–Arabia–Middle East region, and with 85% of SSA values within expected error (EE) of ±0.03 (Lyapustin et al., 2021). The global AOD and SSA accuracy analysis of EPIC MAIAC for the entire mission period 2015–2021 will be published elsewhere.

A comparison of spectral dependence of dust absorption ($\log(k_{440})$ vs $\log(k_{670})$) with AERONET showed a very similar
pattern and slope, albeit MAIAC EPIC gave a smaller range of variation. Such good unbiased agreement was observed for dust at 1 km effective height, indicating that the bulk of dust for the majority of dust storms was in the boundary layer. On the contrary, a similar SSA validation for the wildfire smoke over the USA showed a much better agreement with AERONET for the lofted smoke at 4 km effective height (for 15 out of 17 AERONET sites, Lyapustin et al., 2021). For this reason, the current work uses MAIAC EPIC dust retrievals reported for the 1 km height.


## 2.2 Composition retrieval of hematite and goethite

The forward and inversion algorithms for hematite and goethite retrievals are illustrated in Fig. 2. The forward model uses the Maxwell–Garnett effective medium approximation to simulate complex dielectric function of the total column. Although internal mixing is clearly assumption for minerals, as minerals are generally externally mixed—other than iron oxides
(hematite, goethite; Formenti et al., 2014a), ESMs within each particle mode and AERONET retrievals also assume internal mixing, so the retrieval can be directly fitted into ESMs or compared with AERONET data in the future.

The complex dielectric function ($\varepsilon$), the so-called relative permittivity, is equal to the square of the corresponding complex refractive index ($m^2 = (n + ik)^2$), where $\varepsilon_1, \varepsilon_2, \varepsilon_h$ indicate the complex dielectric functions of inclusion 1, 2, and the host, respectively:


$$\varepsilon_1 = \varepsilon_{1,r} + i\varepsilon_{1,i} = (n_1^2 - k_1^2) + i(2n_1k_1), \tag{2}$$

$$\varepsilon_2 = \varepsilon_{2,r} + i\varepsilon_{2,i} = (n_2^2 - k_2^2) + i(2n_2k_2), \tag{3}$$

$$\varepsilon_h = \varepsilon_{h,r} + i\varepsilon_{h,i} = (n_h^2 - k_h^2) + i(2n_hk_h). \tag{4}$$

Here, inclusions 1 and 2 refer to hematite and goethite, respectively, and the host is generally a homogeneous aerosol mixture whose absorption can be neglected ($k_h(\lambda_j) = 0$). This assumes that all absorption in mixed aerosols is attributed to hematite and goethite particles. The complex refractive indices of hematite and goethite were brought by Scanza et al. (2015) and Bedidi and Cervelle (1993), respectively, as depicted in Fig. 1 (Note that in Scanza et al. (2015) the complex refractive





index of hematite originally cited as personal communication with A. H. M. J. Triaud, 2005;
http://www.atm.ox.ac.uk/project/RI/hematite.html). Here, we linearly extrapolated the goethite refractive index to 340 nm in the UV region. The real part of the refractive index of the host ($n_h(\lambda_j)$) is 1.52, 1.52, 1.51, and 1.5 at 340, 388, 443, and 680 nm, respectively, as determined by Di Biagio et al. (2019).

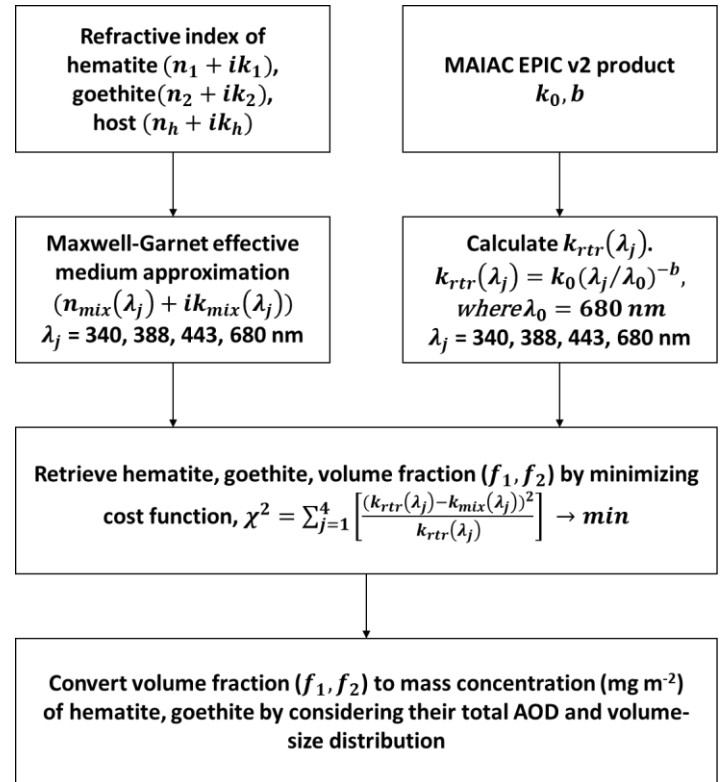

**Figure 2: Schematic diagram of the methodology of this study. Spectral aerosol absorption data from MAIAC EPIC ($b$ and $k_0$ values) are used to infer imaginary refractive indices ($k_{rtr}$) at 340, 388, 443, and 680 nm. These values are fitted with imaginary indices of mixtures with hematite, goethite, and host complex refractive index using the Maxwell–Garnett effective medium approximation rule. The cost functions between the EPIC imaginary and mixture refractive indices are minimized by iteration of volume fractions ($f$) of hematite and goethite (Section 2.2).**

The complex dielectric function of mixed aerosol ($\varepsilon_{MG}$) can be summarized as follows using the Maxwell–Garnett effective medium approximation:

$$\varepsilon_{MG} = \varepsilon_h \left[ 1 + \frac{3\left(f_1 \frac{\varepsilon_1 - \varepsilon_h}{\varepsilon_1 + 2\varepsilon_h} + f_2 \frac{\varepsilon_2 - \varepsilon_h}{\varepsilon_2 + 2\varepsilon_h}\right)}{1 - f_1 \frac{\varepsilon_1 - \varepsilon_h}{\varepsilon_1 + 2\varepsilon_h} - f_2 \frac{\varepsilon_2 - \varepsilon_h}{\varepsilon_2 + 2\varepsilon_h}} \right] = \varepsilon_{MG,r} + i\varepsilon_{MG,i} \tag{5}$$



where $f_1$, $f_2$ refer to volume fractions (unitless) of inclusions 1 and 2, respectively. The volume fraction of the host is $(1 - f_1 - f_2)$. Therefore, the complex refractive index of the mixture is a function of two inclusions and one homogeneous host:

$$m_{mix}(\lambda_j) = F\left(f_1, f_2, m_1(\lambda_j), m_2(\lambda_j), n_h(\lambda_j)\right) = n_{mix}(\lambda_j) + ik_{mix}(\lambda_j) \tag{6}$$


where the real and imaginary parts of the aerosol mixture, derived from the relation $\varepsilon_{MG} = m_{mix}{}^2$, are:

$$n_{mix} = \sqrt{\frac{\sqrt{\varepsilon_{MG,r}{}^2 + \varepsilon_{MG,i}{}^2} + \varepsilon_{MG,r}}{2}} \tag{7}$$

$$k_{mix} = \sqrt{\frac{\sqrt{\varepsilon_{MG,r}{}^2 + \varepsilon_{MG,i}{}^2} - \varepsilon_{MG,r}}{2}} \tag{8}$$


The inversion minimizes the following cost function by iteratively updating the volume fraction of inclusions ($[f_1, f_2]$):

$$\chi^2 = \sum_{j=1}^{4} \left[ \frac{(k_{rtr}(\lambda_j) - k_{mix}(\lambda_j))^2}{k_{rtr}(\lambda_j)} \right] \rightarrow min, \tag{9}$$

where

$$k_{rtr}(\lambda_j) = k_0 (\lambda_j/\lambda_0)^{-b}, where\ \lambda_0 = 680nm. \tag{10}$$

The parameter $\lambda_j$ represents the four EPIC wavelengths of 340, 388, 443, and 680 nm. Unlike Schuster et al. (2016), we minimize only the imaginary term of the mixture refractive index. However, the real parts of the refractive indices of inclusions 1, 2, and the host are significant, as the imaginary term of the mixture ($k_{mix}(\lambda_j)$) is calculated from both real ($\varepsilon_{MG,r}$) and imaginary ($\varepsilon_{MG,i}$) parts of the complex dielectric function. Therefore, realistic values of the real part of the refractive indices

of inclusions 1, 2, and the host are still required as well as those of the imaginary part. Di Biagio et al. (2019) concluded that the real part of the refractive index is generally source- and wavelength-independent with a range of 1.48–1.55 and a sample mean of 1.52, based on a study of 19 mineral dust and soil samples from different main global dust sources. As we mentioned above, we used the values 1.52, 1.52, 1.51, 1.5 at 340, 388, 443, 680 nm, respectively.

The retrieved volume fractions of hematite and goethite ($[f_1, f_2]$), can be converted to mass concentrations (mg m$^{-2}$)

by considering their total AOD and volume–size distributions. Specifically, total AOD ($\tau^a$) is a summation of the fine-mode



and coarse-mode AOD, with MAIAC considering fixed $h_i(\lambda)$ (AOD per unit volume concentration) values and $C_{Vi}$ (volume concentration) for each mode (Lyapustin et al., 2011):

$$\tau^a = \tau_f^a + \tau_c^a = C_{Vf}h_f + C_{Vc}h_c \simeq C_{Vc}h_c \tag{11}$$

For dust, MAIAC uses a dynamic model of size distribution for the Solar Village AERONET site (Dubovik et al., 2002). In this model, the volume fraction of the coarse mode rapidly increases with AOD. This justifies the approximation in (11) as MAIAC provides flexible (AOD, $k_0$, $b$) retrievals only when the background model AOD is high (AOD>0.6). A constant value of $h_c(443) = 1.2526$ for dust yields the following simple equation:

$$C_{Vc} = \tau^a/1.2526 \tag{12}$$

The volume concentration of hematite ($C_{V,hema}$) can be obtained by multiplying the total volume concentration and retrieved volume fraction of hematite:

$$C_{V,hema} = C_{Vc} \times f_{hema} = \tau^a/1.2526 \times f_{hema} \tag{13}$$

and the mass concentration of hematite ($C_{M,hema}$) is calculated by multiplying the corresponding densities ($\rho_{hema}$ = 5260kg m$^{-3}$; $\rho_{goethite}$ = 3800 kg m$^{-3}$; $\rho_{host}$ = 2500 kg m$^{-3}$; Scanza et al., 2015):

$$C_{M,hema} = C_{V,hema} \times \rho_{hema} = \tau^a/1.2526 \times f_{hema} \times \rho_{hema} \tag{14}$$

Accordingly, the mass concentration of goethite ($C_{M,goet}$) and host ($C_{M,host}$) can be calculated by replacing $f_{hema}$ and $\rho_{hema}$ in equation (13) and (14) with the corresponding volume fraction and density. Density of free iron is roughly twice that of other minerals (Schuster et al., 2016; Formenti et al., 2014a).


### 2.3 Hematite refractive index

The published values of laboratory-based spectral refractive indices of hematite in the 300-700 nm range are highly variable (Fig. 3; Table 1; Schuster et al., 2016; Zhang et al., 2015). Hematite is a uniaxial crystal with hexagonal structure. The optical functions of hematite (complex dielectric function ($\hat{\epsilon}$) or complex refractive index ($m$)) have been obtained in laboratory
studies using techniques such as ellipsometry spectroscopy, which provides complex dielectric functions as a function of photon energy of ordinary and extraordinary rays (Chen and Cahan, 1981; Vernon 1962), reflectance spectra measurements





(Querry, 1985; Sokolik and Toon, 1999; Bedidi and Cervelle, 1993), transmission and scattering measurements (Kerker et al., 1979; Hsu and Matijevic, 1985), diffuse reflectance measurements (Gillespie and Lindberg, 1992), and absorption coefficient measurements (Marusak et al., 1980). The refractive index of hematite also has been calculated by combining existing four

measurements of Galuza et al. (1979), Kerker et al. (1979), Steyer (1974) and Onari et al. (1977) (Longtin et al., 1988).

Information on the complex refractive index of goethite is much more scarce, with two types of indices having been published by Bedidi and Cervelle (1993) and Glotch and Roman (2009) for 0.45–0.75 µm and 8–50 µm wavelength ranges, respectively. Hematite and goethite complex refractive indices (or dielectric functions) are needed for the UV–Vis region as *a priori* information for determination of the retrieved volume fraction of hematite and goethite by EPIC MAIAC (Section 2.2).

We therefore examined how the volume fractions of hematite and goethite change with the different types of hematite refractive indices (Sections 2.3 and 4).

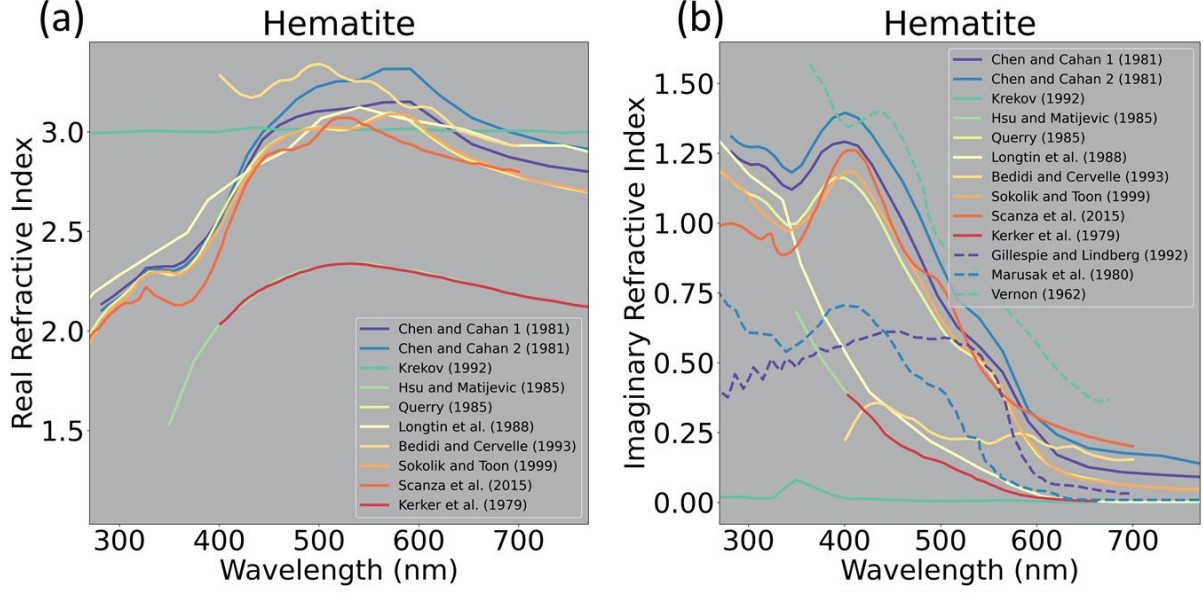

**Figure 3: Plots of previously published hematite refractive indices at 300–700 nm: (a) real part; (b) imaginary part . The reference information is also summarized in Table 1.**




**Table 1: Summary of hematite refractive index information used in Fig. 3.**

| No. | Reference | Wavelength | Structure | Temperature | Method |
|---|---|---|---|---|---|
| 1 | Chen and Cahan (1981) –1* | 263–770 nm | Polycrystalline ($\alpha$–$Fe_2O_3$) | 1100°C | Ellipsometry spectroscopy (* the paper provided two samples sintered at different temperature) |
| 2 | Chen and Cahan (1981) –2* | | | 1250°C | |
| 3 | Krekov (1992) | 200–4500 nm | - | - | - |
| 4 | Gillespie and Lindberg (1992) | 250–700 nm | $Fe_2O_3$ | - | Measure diffuse reflectance, using the dilution method of the Kubelka–Munk (K–M) diffuse reflectance theory |
| 5 | Hsu and Matijevic (1985) | 350–650 nm | Colloidal hematite (0.10, 0.12, 0.13, 0.15, 0.16, 0.51 $\mu m$) | 25°C | Taken from Kerker et al. (1979) |
| 6 | Querry (1985) | 210–900 nm | Uniaxial Crystal ($\alpha$–$Fe_2O_3$) | - | Spectrophotometer, using Kramers–Kronig |
| 7 | Longtin et al. (1988) | 200–300000 nm | $Fe_2O_3$ | - | Combining 4 measurements |
| 8 | Bedidi and Cervelle (1993) | 350–750 nm | Hematite (Ordinary index) | | Koenigsberger formula |
| 9 | Sokolik and Toon (1999) | 200–50000 nm | - | - | Taken from Querry (1985) (Based on clarification by Zhang et al. (2015)) |
| 10 | Kerker et al. (1979) | 400–880 nm (n) 400–700 nm (k) | $\alpha$–$Fe_2O_3$ | - | n: Literature values of the average k: Measure transmission and scattered intensities, using Lorent–Mie theory of scattering by isotropic homogeneous spheres |
| 11 | Marusak et al. (1980) | 207–945 nm | $\alpha$–$Fe_2O_3$ | 298K | Measure absorption coefficient |
| 12 | Vernon (1962) | 350–700 nm | Single–crystal ($\alpha$–$Fe_2O_3$) | - | Ellipsometry spectroscopy |
| 13 | Scanza et al. (2015) | 100–1000000 nm | Hematite | - | http://www.atm.ox.ac.uk/project/RI/hematite.html, cited as personal communication with A. H. M. J. Triaud, 2005 |





## 3 Results

Ginoux et al. (2012) described eight different global dust-source regions. We further divided the North Africa dust
source into two regions, northern Africa and the Sahel, following Di Biagio et al. (2017), and therefore considered a total of
nine different subcontinental dust-source regions (Fig. 4) including northern Africa (region 1), the Sahel (2), eastern Africa
and the Middle East (3), central Asia (4), eastern Asia (5), North America (6), South America (7), southern Africa (8), and
Australia (9). The MAIAC EPIC currently does not retrieve dust aerosol over regions 7 and 8. For regions 1–6 and 9, we
selected 24 significant dust events during 2015–2020 (Table 2; Fig. 5; Figs A1–A6; Fig. S1) for this study.


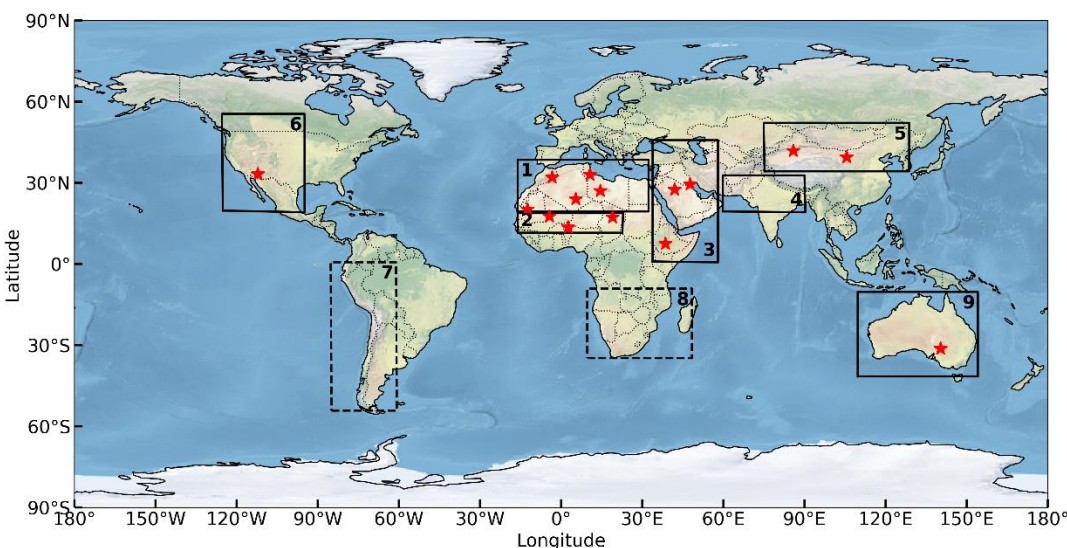

**Figure 4: The nine main global dust source regions: (1) northern Africa, (2) the Sahel, (3) East Africa and the Middle East, (4) central Asia, (5) East Asia, (6) North America, (7) South America, (8) southern Africa, and (9) Australia. From the eight regions originally selected by Ginoux et al. (2012) North Africa was divided in northern Africa and the Sahel (Di Biagio et al., 2017). The**
**MAIAC EPIC version 2 does not provide dust aerosol data over regions 7 and 8 (square box of dashed line). The 15 red stars denote soil locations of Di Biagio et al. (2019) where the soil properties were analyzed for the iron oxides content, complex refractive indices, and SSA.**



**Table 2: Summary of information for selected dust episodes used in this study (Fig. 5; Figs A1–A6). Corresponding aerosol plume height (based on CALIOP) and volume size distribution (based on AERONET) are additionally presented in Fig. S1.**

| Case Num. | Geographical area | Event | Longitude: min, max | Latitude: min, max | Region |
|---|---|---|---|---|---|
| **Case 1** | Sahara–Sahel | January 1, 2018 | −20.00, 46.00 | −20.00, 43.00 | Bodélé Depression |
| **Case 2** | | March 30, 2018 | −20.00, 46.00 | −20.00, 43.00 | Vast Dust over Sahara |
| **Case 3** | | May 30, 2018 | −20.00, 46.00 | −20.00, 43.00 | western Africa to Atlantic Ocean |
| **Case 4** | | February 21, 2016 | −13.35, −2.03 | 32.56, 42.33 | Sahara to Portugal and Spain |
| **Case 5** | | February 22, 2017 | −5.90, 7.94 | 32.27, 41.27 | Algeria to Spain |
| **Case 6** | | August 6, 2015 | 22.32, 36.95 | 13.53, 27.39 | Sudan toward southern Egypt |
| **Case 7** | | August 7, 2015 | 22.32, 36.95 | 13.53, 27.39 | Sudan–Egypt–Red Sea |
| **Case 8** | Middle East | September 1, 2015 | 39.69, 51.01 | 27.74, 37.40 | Middle East (Iraq and Iran), Shamal, Haboob |
| **Case 9** | | September 2, 2015 | 39.69, 51.01 | 27.74, 37.40 | Middle East (Iraq and Iran), Shamal, Haboob |
| **Case 10** | | September 7, 2015 | 34.45, 44.19 | 30.30, 37.93 | Middle East (Iraq and Iran) |
| **Case 11** | | July 28, 2018 | 41.69, 60.15 | 12.58, 25.56 | Saudi Arabia |
| **Case 12** | India | May 3, 2018 | 62.00, 85.00 | 15.78, 35.59 | India dust in Spring (pre–monsoon) |
| **Case 13** | | May 14, 2018 | 62.00, 85.00 | 15.78, 35.59 | India dust in Spring (pre–monsoon) |
| **Case 14** | | June 14, 2018 | 62.00, 85.00 | 15.78, 35.59 | India dust in Late Spring (pre–monsoon) |
| **Case 15** | East Asia | April 30, 2016 | 76.35, 90.59 | 30.96, 46.34 | Taklimakan |
| **Case 16** | | May 1, 2016 | 76.35, 90.59 | 30.96, 46.34 | Taklimakan |
| **Case 17** | | May 4, 2017 | 112.25, 121.35 | 35.72, 45.17 | Gobi Desert to Russia |
| **Case 18** | North America | March 31, 2017 | −109.04, −103.05 | 29.54, 35.94 | Mexico–New Mexico |
| **Case 19** | | March 31, 2017 | −109.04, −103.05 | 29.54, 35.94 | Mexico–New Mexico |
| **Case 20** | | April 10, 2019 | −108.66, −100.96 | 30.44, 37.6 | Mexico–New Mexico |
| **Case 21** | | April 10, 2019 | −108.66, −100.96 | 30.44, 37.6 | Mexico–New Mexico |
| **Case 22** | Australia | February 12, 2019 | 132.96, 153.70 | −38.67, −20.11 | Australia |
| **Case 23** | | February 12, 2019 | 132.96, 153.70 | −38.67, −20.11 | Australia |
| **Case 24** | | February 13, 2019 | 132.96, 153.70 | −38.67, −20.11 | Australia |




## 3.1 Case studies

### 3.1.1 *Northern Africa (Sahara) and the Sahel*

Three different dust events over the Sahara and Sahel regions are depicted in Fig. 5, which are the same episodes illustrated by Lyapustin et al. (2021) in their Fig. 4. In winter (late November to mid-March), western Africa is known for its
northeasterly dry, dusty wind, the "Harmattan". In summer, due to the Intertropical Convergence Zone (ITCZ) moving northward by ~20°N during July–August and returning south by January (close to equatorial over western Africa; Nicholson, 2018), there is a southwesterly humid monsoon flow from the Gulf of Guinea over the low West African Sahelian area (Formenti et al., 2011b).

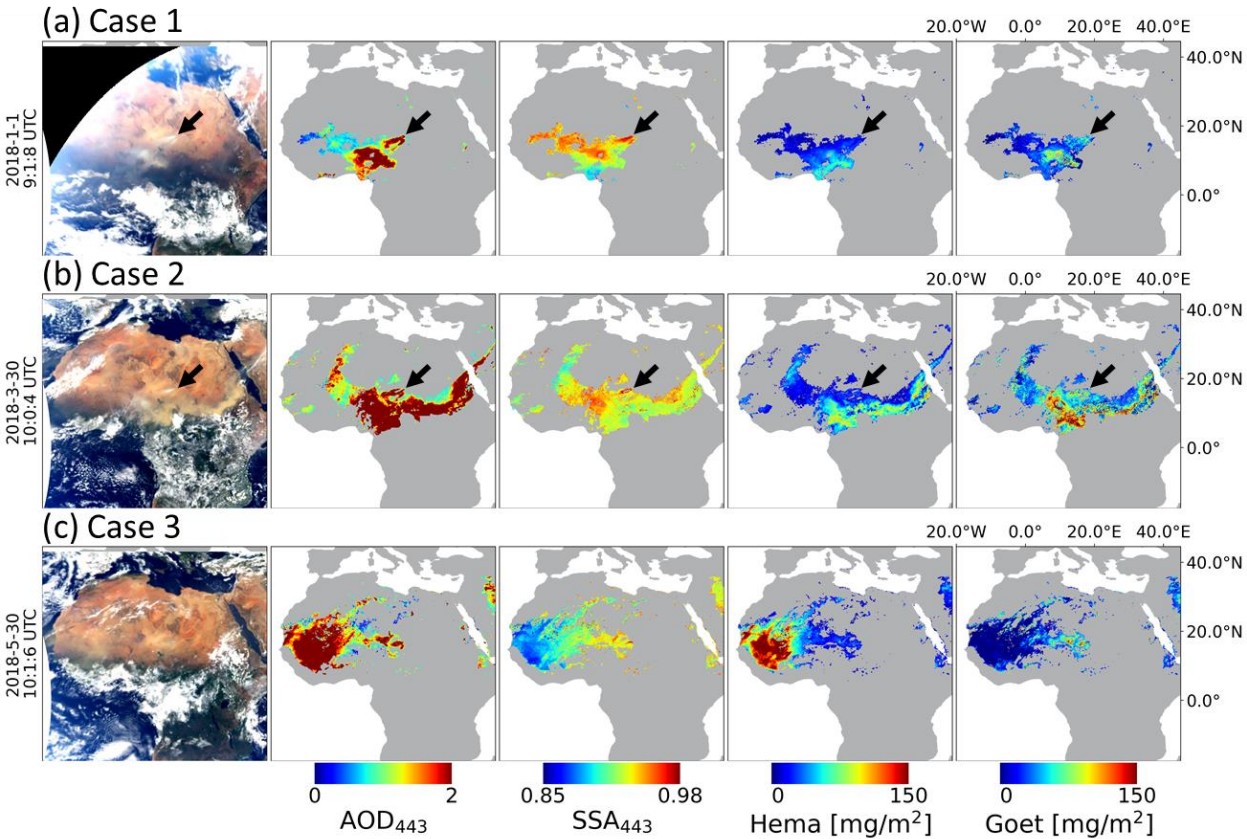


**Figure 5: Dust episodes over Sahara–Sahel region. From left, RGB TOA image generated from EPIC L1B version 3 data, MAIAC AOD and SSA at 443 nm, hematite and goethite mass concentrations for January 1, 2018 (top); March 30, 2018 (middle); May 30, 2018 (bottom). The black arrows point at the Bodélé Depression area.**





The topmost row in Figure 5 shows the Bodélé Depression episode of January 1, 2018. In Chad, this is known as the largest single dust source with the low iron-oxide content (~0.7 wt.%; Di Biagio et al., 2019) due to the diatomaceous sediments of the region (Todd et al., 2007; Moskowitz et al., 2016). On that particular day, the $AOD_{443}$ was high at ~2.0 over the Bodélé Depression; there was negligible hematite mass concentration and up to 75 mg m$^{-2}$ goethite mass concentration, corresponding to 0.5–1 wt.% iron-oxide near the source regions by our calculation (not shown), consistent with the 0.7 wt.% of Di Biagio et

al. (2019). Immediately south of Bodélé, a different dust source was active with higher hematite concentration judging by its reddish color compared with the white dust from the Bodélé Depression.

Second row in Figure 5 illustrates another dust event of March 30, 2018, with typical values of dust 443 nm SSA of ~0.92. This event began on March 28 near the central Sahara and spread to the Sahara and Sahel regions where it persisted for about a week, moving south before spreading to the southwest. During these events, our retrieval shows a variable hematite

concentration of 75–100 mg m$^{-2}$, with higher goethite mass concentrations of 75–150 mg m$^{-2}$ in some areas. This trend towards goethite predominance over hematite in the Sahara and Sahel regions is consistent with the findings of Lafon et al. (2006) and Formenti et al. (2014a), although our results pertain to late March while the former and latter other studies were laboratory generated from soil and in summer or winter, respectively. In the analysis by Formenti et al. (2014a), the goethite content of iron oxides was 52–78 wt.%, with the highest values being for dust originating in the Sahel. Our analysis is showing a similar

range (over 50 wt.%) of goethite in iron oxides in this event.

The third row in Fig. 5 shows dust events with high hematite contents and highly absorbing ($SSA_{443}$ ~0.87) properties over the western Sahara on May 30, 2018. Visual analysis indicates that the dust originated in the western Sahara over Algeria and Mali on May 27, and was transported by northeasterly winds towards the western Sahel, and then on to the Atlantic Ocean.

Another dust case over northern Africa is shown in Fig. A1, with the top row (February 21, 2016) depicting dust

blown northward from the Sahara to Portugal and Spain by the Calima, a warm southeasterly wind common in North Africa in winter. On the same day, AERONET measurements at the Valladolid site in Spain detected a coarse-mode aerosol with $SSA_{443}$ of ~0.875 to ~0.925 (Fig. S1), similar to the MAIAC EPIC SSA recorded over Spain. The second row (February 22, 2017) also depicts dust events involving Calima transport from Algeria to Spain. On that particular day, AERONET at the Granada site in Spain detected the coarse-mode size distribution with $SSA_{443}$ of ~0.90, similar to the MAIAC-retrieved SSA

value of ~0.90. The Calima often carries dust to the Canary Islands, although the atmospheric low-pressure systems may deflect the winds northward. On February 22, 2017, the Cloud-Aerosol Lidar with Orthogonal Polarization (CALIOP) detected dust aerosol mostly at 2–4 km (Fig. S1) above southern Spain. Both the first- and second-row cases in Fig. A1 involved the Calima wind and demonstrated hematite transport to the Iberian Peninsula, whereas goethite transport was negligible.

The third (August 6, 2015) and fourth (August 7, 2015) rows of Fig. A1 describe dust transport north from the Sudan

toward southern Egypt near Aswan, which is clearly visible in red-green-blue (RGB) images generated with TOA reflectance. On both these days, the SSA value of the plume as determined by MAIAC was slightly below 0.92, the typical dust SSA value. Ginoux et al. (2012) described this area as the "Nile River Basin", containing essentially natural sources. Prospero et al. (2002)





determined that dust sources over Egypt become active during March–October. Our retrieval results indicate that the iron oxides there comprised mainly hematite with a spatial distribution similar to that of the plume $AOD_{443}$.

### 345    3.1.2 Middle East

Selected dust cases over the Middle East are shown in Fig. A2, where the top two rows (September 1 and 2, 2015) represent the general characteristics of the Shamal, with hot and dry northwesterly winds blowing towards the Persian Gulf in summer, including the Haboob wind. This dust event was triggered on August 31 by a surface low-pressure system on the Syria–Iraq border, moving across Iraq, Iran, and the Persian Gulf region over the next two to three days. The first row
(September 1) in Fig. A2 depicts a cyclone-shaped dust storm with $AOD_{443} > 2$ and a moderately absorbing $SSA_{443}$ value (~0.92). Hematite was located more toward the center of the cyclone, whereas goethite was distributed within its tail, toward the east-southern part of the cyclone core. On September 2 (Fig. A2, second row), the dust storm moved towards the coastline near the Persian Gulf with consistent $AOD_{443}$ and $SSA_{443}$ values of ~2 and ~0.92, respectively. However, the reduced spectral absorption exponent $b$ with a slightly increased $k_0$ value indicates a change in chemical composition (not shown). The cyclone
appears to have a higher goethite content than hematite in its core. Note that both first row and second row in Fig. A2 are observed at 6:50 UTC and 6:59 UTC, respectively. Therefore, the changed chemical composition are not likely affected by geometry dependency of the MAIAC EPIC ($b$, $k_0$) retrieval algorithm.

Fig. A2, third row (September 7, 2015), shows the dust first emerging on September 6 near the Euphrates River in Syria. By September 7 the dust had engulfed Syria, northern Jordan, and southern Turkey over the Mediterranean coast. On
September 7, the Sede Boker AERONET site in southern Israel detected a coarse-mode dominant volume–size distribution and corresponding SSA of 0.89–0.92 (Fig. S1). By September 9, the dust had moved southwest over northeastern Egypt, and Sede Boker AERONET site detected $AOD_{443} > 2.75$. Ginoux et al. (2012) described this source region as comprising 'anthropogenic', 'natural', and 'hydrologic' aerosol, and named it the 'Mesopotamia source', which is most active during June–August. Ginoux et al. (2012) considered it 'natural' if <30% of the area was agricultural; otherwise it was 'anthropogenic'
(>30% agricultural). If the dust-event incidence is associated with ephemeral water bodies, the source is termed 'hydrologic'. For the Euphrates River area, farmland northeast of the city of Ar Raqqah in Syria has the most frequently active anthropogenic dust sources (Walker et al., 2009), and several salt-flat areas near the Iraq–Syria border are significant natural dust sources (Ginoux et al., 2012). Our retrieval results indicate that on September 6, hematite was predominant northeast of Ar Raqqah, but on September 7 (third row, Fig. A2) the goethite content was predominant, likely owing to different land use (Fitzpatrick,
2004; Journet et al., 2014).

The Fig. A2, fourth-row, episode (July 28, 2018) describes another dust event in the Shamal over the Rub'al Khali sandy desert area. This event swept across the Arabian Peninsula during July 27–28, with a swirling cyclone shape over part of the Persian Gulf, Gulf of Oman, Arabian Sea, and Gulf of Aden during July 29–30. Near this dust-storm region, there are two dust sources (Ginoux et al., 2012), the 'Empty Quarter' and 'highlands of Saudi Arabia'. The former refers to a salt-flat





region named 'Sabkha Matti' on the border between the United Arab Emirates and Saudi Arabia. The latter lies in the northern
Rub'al Khali sandy desert area, which comprises three dry riverbeds (Ginoux et al., 2012). Our chemical-composition retrieval
results (Fig. A2) indicate a hematite-dominated distribution overall but mixed with goethite along the coast of Oman.

### 3.1.3 Central Asia (India)

A dust event over India is depicted in Fig. A3. Large amounts of dust were observed during the pre-monsoon season
of March–May over northern India, with the lowest dust incidence in the post-monsoon season of September–November
(Ginoux et al., 2012). Pre-monsoonal dust storms are attributed primarily to the Arabian Peninsula and the Thar desert, with
wind transport to northwestern India (Sarkar et al., 2019). Three 2018 dust storms (Fig. A3) swept across the Rajasthan and
Uttar Pradesh regions by westerly or southwesterly winds from western India. The Himalayan mountain region blocks dust
movement further north and usually deflects transport eastward.

Fig. A3, row one (May 3, 2018) indicates dust with $AOD_{443}$ values of 1–2 near Rajasthan with an $SSA_{443}$ value of
~0.92. Despite the uniform SSA over the aerosol plume, our compositional retrieval results indicate that hematite was
predominant in the northern part of the plume, with goethite occurring only in that part over the Thar Desert near to the border
with Pakistan or over coastal areas of western India.

One week later (row two, May 14, 2018) and a month later (row three, June 14), the same Rajasthan region suffered
severe dust storms again, with these being recorded as anomalously intense dust storms. In the latter event, dust was trapped
over northern India by southerly winds from the Bay of Bengal. In both cases, hematite was concentrated more towards the
center of the plume, with concentrations of >100 mg m$^{-2}$, whereas the goethite content was small (<50 mg m$^{-2}$).

The Kanpur region of northern India may have aerosol properties reflecting mixtures of pollution aerosol from the
combustion of fossil fuels and biofuels, and desert dust (Eck et al., 2010). During the events depicted in Fig. A3, the AERONET
inversion data (version 3, level 2) for Karachi (northwestern coast of India) indicated coarse-mode dominant size distribution
on May 3 and 14 (rows two and three, Fig. A3) with SSA values of ~0.9 (Fig. S1), whereas in the Kanpur region (northern
India) small bio-modal size distributions were detected (fine/coarse volume–size distribution peak 0.04/0.18 on May 3;
0.12/0.42 on May 13; 0.05/0.46 on June 14). This implies that dust aerosol might have been mixed with fine-mode pollution
aerosol over northern India. Our retrieval algorithm currently fits only the pure dust model, so the retrieved hematite and
goethite concentrations may be biased in this case.

### 3.1.4 East Asia

In East Asia, there are two main regional sources of mineral dust, the Taklimakan and Gobi deserts. Fig. A4 (first and
second rows) illustrates a Taklimakan Desert dust event in spring (April 30 and May 1, 2016). During this event, dust was
transported by easterly winds and circulated clockwise around the Tarim Basin for two days. The Tarim Basin is flanked by
~5-km-high mountain ranges to the west, south, and north; therefore, dust storms usually blow in from the east at low levels,





trapping the event in the basin. Strong surface winds may carry dust to altitudes of ~10 km, transporting it across China and the Pacific Ocean. On April 30, 2016, a severe dust event ($AOD_{443}$ ~2.0) with $SSA_{443}$ values of ~0.90 displayed hematite predominance over goethite at 07:27 UTC. Over the next two days, the $SSA_{443}$ increased to ~0.95 with the reduced hematite content, while goethite content increased slightly to 50–60 mg m$^{-2}$.

Fig. A4, row three, describes a dust event originating in the Gobi Desert with dust being transported to Russia over East Asia. This event was pushed by a cold front over the Gobi Desert near the border of China and Mongolia and was clearly captured by MODIS Aqua on May 3, 2017 (not shown). On May 4, the dust moved to the Russian border just east of Mongolia in a cyclonic circulation, bypassing the Beijing area. The AERONET sites in Beijing (Fig. S1), Beijing_RADI, and XiangHe detected coarse-mode dominant volume–size distributions with SSA values of ~0.91 on May 4, implying pure dust aerosol.
Our algorithm retrieved moderate goethite concentrations (~75 mg m$^{-2}$) with low hematite concentrations (<50 mg m$^{-2}$).

### 3.1.5 *North America*

Dust events over North America near the White Sands in New Mexico are illustrated in Fig. A5, for March 31, 2017 (first and second rows) and April 10, 2019 (third and fourth rows). The highest frequencies of dust event over North America occur in the southwestern USA and northern Mexico at up to 30% of the time in spring (Ginoux et al., 2012). However,
MAIAC EPIC version 2 does not retrieve dust aerosol over the southwestern USA, so only northern Mexico events are considered here.

On March 31, 2017 (first and second rows, Fig. A5), MAIAC captured dust patterns changing hourly in the southern White Sands area. The likely dust source was the Chihuahuan Desert near the border of Mexico and New Mexico, south of El Paso International Airport or near Janos in Mexico (based on MODIS RGB images not shown here). In both the March 31
cases, the goethite concentration was not much (<50 mg m$^{-2}$). Hematite was also rare at 19:05 UTC (first row) with $AOD_{443}$ < 1.0 and $SSA_{443}$ ~0.91, but reached 50–75 mg m$^{-2}$ at 20:53 UTC (second row), increasing to $AOD_{443}$ > 1.5 and $SSA_{443}$ < 0.91.

The April 10, 2019, dust storm was induced by a low-pressure meteorological system. Winds lofted dust from several sources around the region, including the White Sands area, Lordsburg Playa in southwestern New Mexico, the Chihuahuan Desert, and near the border of Mexico and New Mexico. In Fig. A5 (third and fourth rows), the white circle of the White Sands is located near the center of the RGB image. Dust storms passed to the southeast of that area with moderate hematite contents
(50–75 mg m$^{-2}$) and small goethite contents (<50 mg m$^{-2}$).

### 3.1.6 *Australia*

Australia often referred to as the "red continent" (Viscarra Rossel et al., 2010), with highly absorbing dust aerosols (Di Biagio et al., 2019). In Fig. A6, the first (February 12, 2019; 03:44 UTC) and second (05:32 UTC) rows depict a dust event
near Lake Eyre and Lake Blanche, both salt lakes in central South Australia—an area considered the 'Lake Eyre Basin', which is the most active dust source in Australia (Ginoux et al., 2012). Hourly EPIC RGB images indicate that the event started on





February 12 at 01:56–03:44 UTC (not shown). On that day, the AERONET sites at Birdsville (25.9°S, 139.3°E), in the northwest of the dust plume, detected coarse-mode-dominant volume–size distribution with $SSA_{443}$ ~0.82 at 07:23 UTC (Fig. S1). Our retrieval results ($SSA_{443}<0.85$) indicate high hematite concentrations, attributable to the iron-rich soils in the region.

On February 13 (row three, Fig. A6), the dust likely left-over from the previous day kept moving by southwesterly wind. The CALIOP instrument detected dust aerosol over eastern Australia at 32°–26°S at 0–3 km height (Fig. S1). The northern part of the dust plume might had passed the source of Simpson Desert, where dust activity is most frequent in the austral spring and summer (Ginoux et al., 2012). Overall, hematite was predominant over the area with $AOD_{443} > 1.5$.

### 3.1.7 Summary plot of the case studies

The 24 cases described here are summarized in Fig. 6. Over the Sahara, Sahel, and the Middle East, the iron-oxide volume fractions are comparable with those over North America, but the high $AOD_{443}$ values yield higher mass concentrations than those in North America. Although Australia had lower $AOD_{443}$ values than those of Sahara, Sahel and Middle East, the high volume fraction equated with high mass concentrations of hematite. The mass concentration of iron-oxide was relatively low over India, East Asia, and North America. Fig. 7 shows the relationship between iron-oxide mass fraction, $SSA_{443}$, and imaginary refractive index at 443 nm. The previous study of Di Biagio et al. (2019) and Moosmüller et al. (2012) dealt with this relationship to estimate the spectrally resolved SW absorption of dust based on composition. Our retrievals were consistent with results of Di Biagio et al. (2019) in the relationship between iron-oxide mass fraction and $SSA_{443}$, but showed discrepancy in the relationship between iron-oxide mass fraction and imaginary refractive index at 443 nm likely due to the use of spherical particle shape in Di Biagio et al. (2019) calculation of imaginary refractive index (Kalashnikova and Sokolik, 2004).

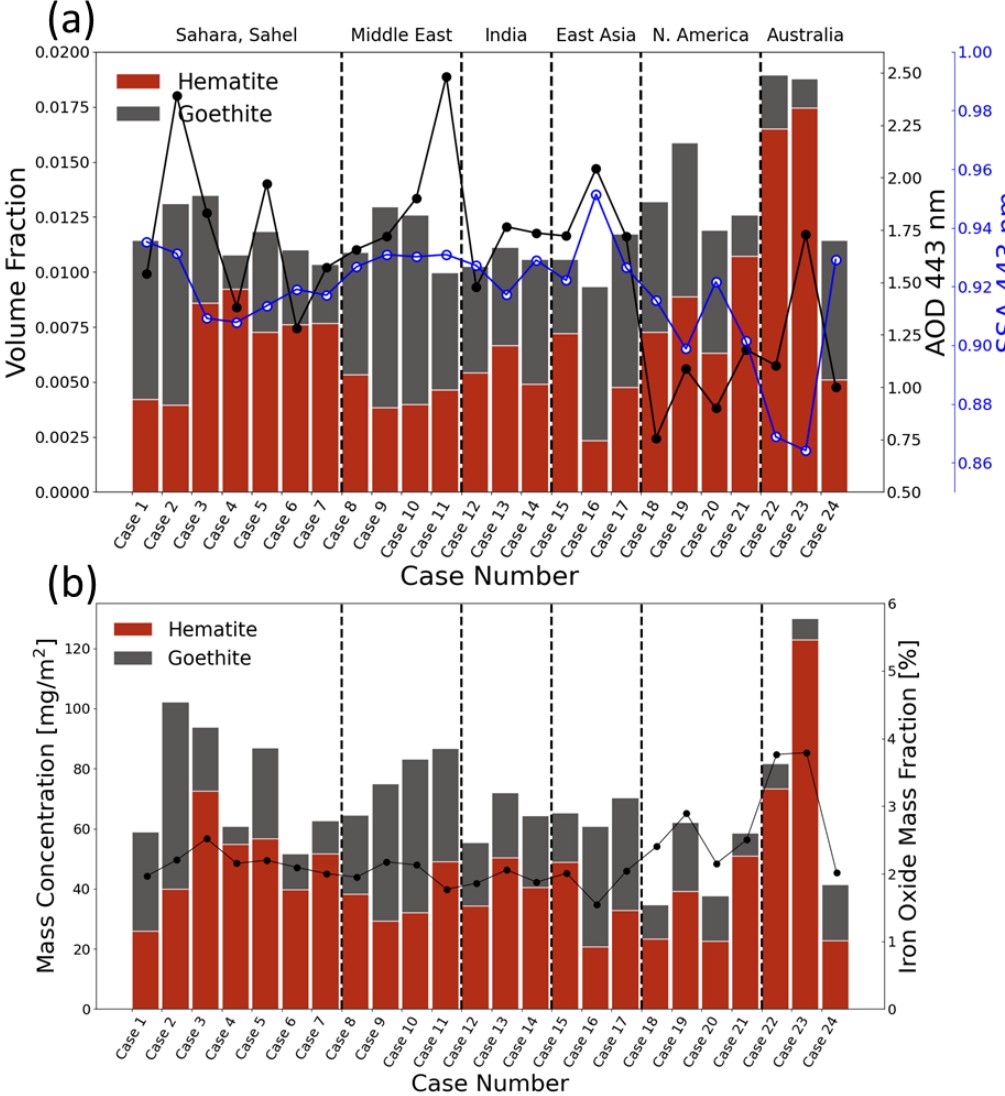


**Figure 6: Bar plot for the 24 cases described in Section 3.1. Case numbers for each row from Fig. 5, Figs A1–A6 are shown on the x axis and summarized in Table 2. (a) Mean volume fractions of hematite (red) and goethite (dark gray). Black and blue points show the corresponding mean AOD (right axis) and SSA at 443nm for each case. (b) Mean mass concentrations of hematite (red) and goethite (dark gray). Black points indicate corresponding iron-oxide mass fraction (wt.%; right axis) of total dust concentration.**






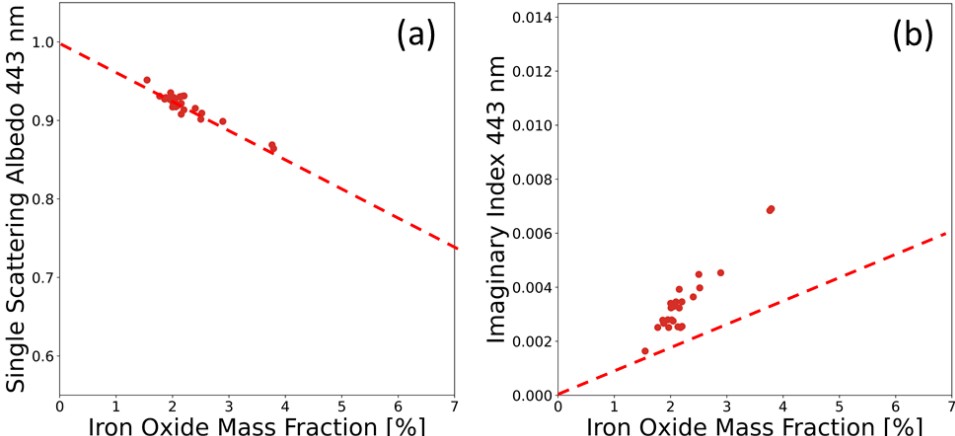

**Figure 7: Relationship between iron-oxide mass fraction (wt.%; x axis) and SSA 443 nm (y axis, (a)) and imaginary refractive index (y axis, (b)). Red points represent the mean values of the 24 cases described in Section 3.1. Dashed line shows the estimated relationship at 443 nm from Di Biagio et al. (2019).**


## 3.2 Comparison with *in situ* measurements of soil samples

Di Biagio et al. (2019) collected soil samples from 19 main dust sources representing global dust optical properties and determined their hematite and goethite contents by the X-ray absorption near-edge structure (XANES) method, providing bulk compositions of pure dust aerosol in dry conditions with a size range of 2–6 days of transport following emission. We compare
their measured soil hematite and goethite mass concentrations (wt.%) with our retrieval results over 15 sites in Fig. 8 (for statistics, see Table S1). Only satellite-data pixels with $AOD_{443} > 1.0$ were used in the monthly composite, as some pixels of goethite retrieval with $AOD_{443} < 1.0$ display "noise" or "blob" patterns (e.g., third and fourth rows, Fig. A1). This noise may be attributed to their low signal to noise ratio with low AOD.

In Fig. 8, Tunisia (33.02°N, 10.67°E) and Morocco (31.97°N, 3.28W) represent northern Africa, Libya (27.01°N,
14.50°E) and Algeria (23.95°N, 5.47°E), represent the central Sahara, Mauritania (20.16°N, 12.33°W) represents western Africa above the Sahel latitude (~20°N), Niger (13.52°N, 2.63°E), Mali (17.62°N, 4.29°W) and Bodélé (17.23°N, 19.03°E) represent the Sahel, Ethiopia (7.50°N, 38.65°E) represents eastern Africa, Saudi Arabia (27.49°N, 41.98°E) and Kuwait (29.42°N, 47.69°E) represent the Middle East (especially the Arabian Peninsula), Gobi (39.43°N, 105.67°E) and Taklimakan (41.83°N, 85.88°E) represent East Asia, Arizona (33.15°N, 112.08°W) represents North America, and Australia (31.33°S,
140.33°E) is represented by sites near Lake Eyre in central South Australia. The locations of all of the above are shown in Fig. 4.

**Figure 8: Monthly variations (5ᵗʰ, 50ᵗʰ, and 95ᵗʰ percentiles) of hematite (red shading) and goethite (gray shading) mass concentration (wt.%) calculated from MAIAC EPIC at each site for 2018. Solid lines at the centers of shaded areas indicate median values for hematite (red) and goethite (black). Each site represents the area of ± 1° of MAIAC EPIC data (pixel with AOD > 1.0) except for Australia (±3°). Dashed lines indicate hematite (red) and goethite (black) mass concentrations with ±10% uncertainty (Di Biagio et al., 2019). At the Algeria site, the two dashed lines coincide (Di Biagio et al., 2019). The invisible dashed lines of goethite (black dashed line) at Mauritania, Bodélé, Ethiopia, Kuwait, Gobi, Arizona and Australia indicate zero goethite mass concentration. Di Biagio et al. (2019) does not provide data at Taklimakan site. Monthly variations statistics are additionally presented in Table S1.**






Although Di Biagio et al. (2019) collected data in each origin, the EPIC-retrieved hematite and goethite data may represent both local erosion and transported iron oxides, and these datasets may differ owing to seasonally and spatially varying dust transport from the different sources. Overall, however, our retrieved data tend to cover their entire range of hematite and goethite mass concentrations, except for the Niger site.

**3.2.1. Characteristics over northern Africa and the Sahel – Libya, Algeria, Mauritania, Niger, Mali, and Bodélé**

Sites in the central Sahara and Sahel, such as Libya, Algeria, Mauritania, Niger, Mali, and Bodélé, and in the Middle East such as Saudi Arabia and Kuwait, exhibit greater temporal variability than other sites, implying large dynamic dust-source variability in the Sahara, Sahel, and Middle East.

In the Sahara and Sahel, goethite is usually the predominant iron-oxide species by mass (Di Biagio et al., 2019; Lafon et

al., 2006; Formenti et al., 2014a, b), as also indicated partially by our results. At four sites (Mauritania, Niger, Mali, and Bodélé) in the Sahel, goethite predominated over hematite during August–April, which included the Harmattan period. However, in early summer (May–July), the median hematite value was higher than that of goethite at these four sites, with higher hematite concentrations in the western Sahel peaking in May (Mauritania, ~2.9 wt.% > Niger, ~2 wt.% > Mali, ~1.8 wt.% > Bodélé, ~1.2 wt.%). The soil analysis data of Di Biagio et al. (2019) displayed a similar trend (Mauritania, 3.3 wt.% >

Niger, 2.3 wt.% > Mali, 2.0 wt.% > Bodélé, 0.7 wt.%). Formenti et al. (2011a) and Lazaro et al. (2008) found that hematite is likely to predominate over goethite. Lazaro et al. (2008) detected hematite to goethite ratios of 0.5–2.0 in the Canary Islands, sourced from chotts in Tunisia and Northern Algeria (PSA NAF-1) and foothills of the Atlas Mountains in western Sahara and western Mauritania (PSA NAF-2), and the Mali region. Lazaro et al. (2008) also found hematite to goethite ratios of >1.0 in samples from February, March, May, August, September, and December, and ratios of <1.0 in January, February, June, August,

and September. Schuster et al. (2016) described a maximum volume percentage of iron oxides associated with hematite (%$H$ = $V_{hem}$ / ($V_{hem}$ + $V_{goe}$) × 100) of 83%–93% (median volume percentage of iron oxides associated with hematite of 38%–58%) over western Africa for pure dust cases, with the calculated percentage depending on hematite refractive index. The high-hematite dust regions are thus likely to exist in Mauritania, Niger, Mali in summer.

**3.2.2. Tunisia, Morocco**

In Tunisia, the median values of monthly composite of hematite and goethite are similar to those of Di Biagio et al. (2019; hematite, 1.2 wt.%; goethite, 1.1 wt.%) during January–May and in October, with retrieval values of ~1.2 wt.% for both. Data for the Morocco sites also agree well with those of Di Biagio et al. (2019) in March (hematite, 0.4 wt.%; goethite, 1.0 wt.%). However, the hematite concentration (~2 wt.%) was overestimated relative to soil data during June–August over both Tunisia

and Morocco. Li et al. (2021) mentioned that hematite content in soil climatology J2014 (Journet et al., 2014) exhibits strong regional differences with mass fractions predominantly below 1.5 wt.% but reaching up to 5.0 wt.% in some arid regions such as over northern Africa. Total iron oxides mass concentrations vary over the range about 2–6.5 wt.% globally (2–5 wt.% of



Linke et al. (2006); 2.8–6.5 wt.% of Schuster et al. (2016)), meaning that hematite may be dominant in soil over some parts of northern Africa. Our results for Tunisia and Morocco are thus likely to occur in northern Africa.


### 3.2.3. Niger

Niger, in the central Sahel, lies downwind of some of the most persistent Saharan dust sources, including the Bodélé Depression in Chad, areas in northern Mali and southern Algeria, and also areas of Libya, Egypt, and the Sudan (Formenti et al., 2014a). Our median concentration of hematite (~2 wt.%) in May is consistent with that of Di Biagio et al. (2019; 2.3 wt.%), but goethite was underestimated (<1.8 wt.%; cf. 3.5 wt.% for Di Biagio et al., 2019). The goethite value of Di Biagio et al. (2019) corresponds to our highest value for monthly composite data for January–June (not shown). For Niger, Lafon et al. (2004) found iron-oxide concentrations of $2.8 \pm 0.8$ wt.% during the Harmattan season (November–March), with a major source being Bodélé in the Chad basin, whereas concentrations of $5.0 \pm 0.4$ wt.% were detected during the local erosion season (May–July). Our retrieval was consistent with the 2.8 wt.% value for the Harmattan season, but underestimated the local erosion season value. The Niger soil sites were close to the Banizoumbou AERONET site. MAIAC EPIC retrievals were in good agreement in accuracy with AERONET data for 2018 at this site, with $AOD_{443}$ (R, 0.8; RMSE, 0.23; MBE, -0.11) and $SSA_{443}$ within 0.03, 89% of samples and all within 0.05. Therefore, it is not likely the mixture aerosol type or the MAIAC AOD, SSA retrieval accuracy problem. AERONET direct AOD measurement data at Banizoumbou included only five days of measurement data in June, and most AOD values in July were <0.6 because of the May–October rainy season (with most rain in July–August). Our retrievals might not have detected iron oxides due to the low June–July AOD.

### 3.2.4. Bodélé

The Bodélé area is known to have low iron-oxide concentrations due to the presence of diatomic sediments (Todd et al., 2007; Moskowitz et al., 2016), and the high incidence of high-aerosol events is associated with the Venturi effect of Harmattan winds passing between the Ennedi and Tibesti mountains (Ginoux et al., 2012). The median hematite concentration was consistently low (<1.4 wt.%) over the whole of 2018, consistent with the value (0.7 wt.%) recorded by Di Biagio et al. (2019). Moreover, Moskowitz et al. (2016) concluded that dominant goethite and subordinate hematite together compose about 2 wt.% of iron oxides from the Bodélé Depression, similar trends with our data throughout the year. Iron-oxide levels over Bodélé (<2 wt.%) are shown later in Fig. 9, highlighting different areas especially in winter season, with a consistently high AOD average.


### 3.2.5. Ethiopia

The Ethiopia sites of Di Biagio et al. (2019) are near the Abidjatta–Shalla National Park. We occasionally detected dust over this area in June, with 150 pixels, and with <25 pixels in July. In June, median EPIC hematite value was ~1.2 wt.%, below the 2 wt.% indicated by soil measurement data.






### 3.2.6. Saudi Arabia, Kuwait

The hematite concentration at Saudi Arabia (median ~1.2 wt.% in May) was generally higher than that at Kuwait (~1 wt.% in February and May), consistent with the results of Di Biagio et al. (2019), with 1.8 wt.% in Saudi Arabia and 1.5 wt.% in Kuwait. In Saudi Arabia, the soil goethite content was 0.8 wt.%, similar to our retrieval during March–May (~1 wt.%). Both
sites were dominated by goethite during summer–autumn (July–September). Summer and early autumn is the Shamal season when northwesterly winds blow over the Saudi Arabia peninsula (Yu et al., 2016), with the Saudi Arabia and Kuwait sites (Fig. 8) lying downwind. Therefore, transported dust by the Shamal might have caused the increase in dust goethite content.

### 3.2.7. Taklimakan, Gobi

Over East Asia, the Gobi Desert dust area was active mainly in April in our retrieval (with ~300 pixels), while the Taklimakan was active throughout the spring, early Summer, and September, all with >500 pixels of retrieval. In April, our hematite retrievals were consistent with data of Di Biagio et al. (2019), ~1 wt.% and 0.9 wt.%, respectively. Lafon et al. (2004) measured iron-oxide concentrations of ~3.7 wt.% downwind of Gobi Desert, but our median value even in April seems to underestimate that value. However, our retrievals were generally consistent with the hematite to goethite ratio by mass (~0.55)
found by Shen et al. (2006) who investigated hematite to goethite mass-concentration ratios in bulk samples of eolian dust over Dunhuang (40.3°N, 94.5°E), Yulin (38.2°N, 109.4°E), and Tongliao (43.1°N, 122.1°E), yielding values of 0.57 ± 0.26, 0.59 ± 0.16, and 0.46 ± 0.13, respectively.

### 3.2.8. Arizona and Australia

Both the Arizona and Australia sites may include significant smoke-event effects causing the increase of aerosol absorption that was attributed to the hematite content in our retrieval shown in Fig. 8. However, our case studies over North America (case 18–21) and Australia (case 23) presented in Fig. 6(b) cover the range suggested by Di Biagio et al. (2019) which is 1.5 wt.% (North America) and 3.6 wt.% (Australia) of soil measurement hematite content. Further study is needed to distinguish between smoke and dust in MAIAC EPIC algorithm.

Our retrievals thus generally follow the trends in hematite data recorded by Di Biagio et al. (2019) with some differences, especially for goethite likely associated with transport from different sources over the designated sites.

### 3.3 Global climatology

Seasonal average EPIC data for Sahara–Sahel and Middle East (Fig. 9) and Asia (Fig. 10) are shown for the period January 1 to December 31, 2018 (for global data, see Fig. S2–S5). As in Fig. 8, pixels of AOD > 1.0 were used to compute the average. Currently, MAIAC EPIC cannot differentiate between the smoke and the dust. It is mostly biomass burning smoke in the Indo-Gangetic plane (in Fig. 10) except during the pre-monsoon season, although there is also a dust from Thar desert. Also, part





of the data over North America and Australia (in Fig. S4, Fig. S5) may include some mixed aerosol or smoke cases, and thus
may be biased relative to pure-dust cases (Fig. 5; Figs A1–A6). Therefore, the results in the aforementioned region should not
be considered as accurate. Nevertheless, our results display the generally known iron-oxide patterns.

Globally, our iron-oxide mass concentrations were in the range of up to ~6.5 wt.%, consistent with the generally accepted
range (Di Biagio et al., 2019; Schuster et al., 2016).

The North Africa and Middle East (Fig. 9) areas exhibit increasing AOD throughout by spring and summer (barring the
rainy season over the Sahel near Niger), with levels decreasing after autumn. Hematite, on the other hand, is likely prevalent
over Africa only during May–July, particularly over west Sahara near Mauritania where dust storms occur every year, often
transporting dust across the Atlantic Ocean to Puerto Rico and northern Brazil. Our visual analysis implies that these high dust
hematite concentrations (especially in May) are due to transport from central Algeria, Mali, and sometimes the Sudan, with
transport westward across the Sahel line. Otherwise, goethite was more prevalent than hematite over the Sahara and Sahel
(Formenti et al., 2014a).

The Bodélé Depression area has lower iron-oxide concentration of close to ~1 wt.% with high AOD throughout the year
(Todd et al. 2007; Moskowitz et al., 2016). Higher iron-oxide concentrations are also well documented for the Sahelian area
(0°–20° N) and in the Sahara, particularly in April (not shown). Sahel dust is richer in iron-oxide than Saharan and Chinese
dust, as found by Claquin et al. (1999) and Lafon et al. (2004).

Over Asia (Fig. 10), Taklimakan dust contained twice as much goethite as hematite during March–October (Shen et al.,
2006). The Gobi Desert showed widespread dust activity overall with higher goethite concentrations during March–May, with
less in summer. India had higher hematite concentrations during the pre-monsoon season, especially in May.

Other well-known dust areas, which have not been extensively used as case studies, occur east of the Aral Sea, in the
southeastern coastal region of the Caspian Sea, in eastern Uzbekistan and Turkmenistan, and in the southwestern corner of
Afghanistan, all of which are known highly active dust sources (Ginoux et al., 2012), but knowledge of their dust mineral
composition is lacking.





**Figure 9: Seasonally average data for Sahara–Sahel and Middle East dust source areas generated from MAIAC EPIC 2018 data, (January 1 to December 31): (a) hematite mass concentration (mg m$^{-2}$); (b) goethite mass concentration (mg m$^{-2}$); (c) iron-oxide mass concentration (wt.%), with one row per season (row 1, March, April, and May (MAM); row 2, June, July, and August (JJA); row 3, September, October, and November (SON); row 4, December, January, and February (DJF)). Climatology data may include some mixed aerosol and smoke aerosol cases causing bias relative to pure dust cases (Fig. 5; Figs A1–A6). Four black arrows over (c) are pointing at the Bodélé Depression area. Additional data for AOD at 443 nm, SSA at 443 nm, and total number of datapoints used in this figure are provided in Fig. S2.**



**Figure 10: The same as in Fig. 9 but for Asia. Additional data of AOD at 443 nm, SSA at 443 nm, and total number of datapoints used in this figure are provided in Fig. S3.**


## 4 Implication from different hematite refractive indices

Hematite refractive indices exhibit a large range in the literature, as introduced in section 2.3 and Fig. 3. This is critical since the retrieved hematite and goethite content vary significantly depending on the *a priori* refractive index of hematite. To find the most suitable *a priori* hematite refractive index, we analyzed how much the hematite and goethite content change



quantitatively with respect to our *a priori* hematite refractive index for the 24 dust cases presented in section 3.1. Note that,

some of the literatures do not include real refractive indices, such as Gillespie and Lindberg (1992), Marusak et al. (1980), and

Vernon (1962), so we used the Scanza et al. (2015) real refractive index of hematite as an alternative.

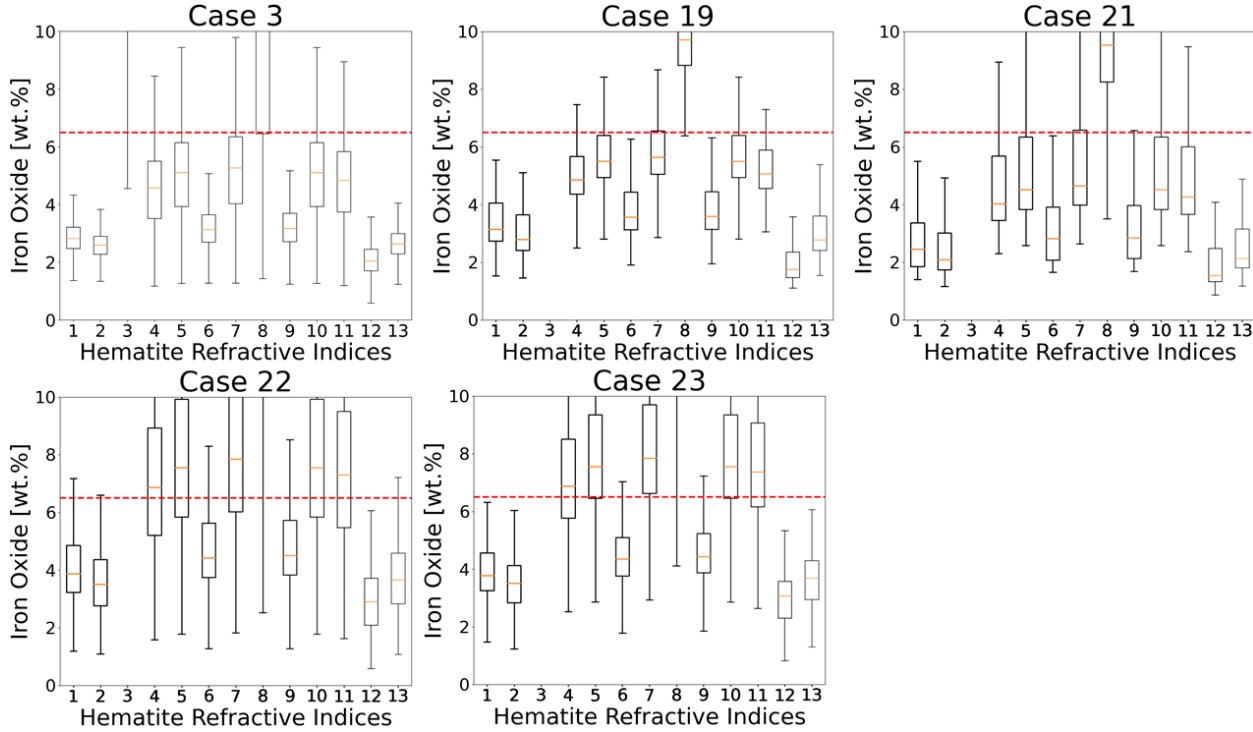

**Figure 11: Box and whisker plot of iron-oxide content by mass (wt.%; y axis) for 13 models of hematite refractive index (x axis;**
**number 1–13, see Table 1) for the dust cases 3, 19, 21, 22, and 23 presented in section 3.1. Red horizontal dashed line on each figure**
**shows the maximum expected iron-oxide content (6.5 wt.%) based on *in situ* measurements.**

Fig. 11 depicts the box and whisker plot of iron-oxide content by mass (wt.%) with respect to 13 kinds of *a priori* hematite

refractive indices (Fig. 3; Table 1) for the selected 5 dust cases (case 3, 19, 21, 22, 23; for all 24 dust cases, see Fig. S6). Note

that the goethite density vary in the literature ($\rho_{goethite} = 3800\ \mathrm{kg\ m^{-3}}$ of Bedidi and Cervelle (1993); $\rho_{goethite} = 4280\ \mathrm{kg\ m^{-3}}$

of Formenti et al. (2014a)), but does not affect the calculated iron-oxide content (wt.%) significantly, as the retrieved volume

fraction of goethite mostly small (less than 0.03) as opposed to over 0.9 volume fraction of host (not shown).

In literature, the *in situ* measurement of maximum of iron-oxide content is about 6.5 wt.% (2–5 wt.% of Linke et al. (2006)

and Formenti et al. (2014b); 2.8–5 wt.% of Lafon et al. (2004); 2.4–4.5 wt.% of Formenti et al. (2008); 0.7–5.8 wt.% of Di

Biagio et al. (2019); 2.8–6.5 wt.% of Lafon et al. (2006)). This indicates that the hematite refractive indices of '3. Krekov

(1992)' and '8. Bedidi and Cervelle (1993)' are not viable for our approach, since the corresponding retrieved iron-oxide

contents mostly exceed 6.5 wt.%. Specifically, the lower quartile for '3. Krekov (1992)' is always over 10 wt.%, therefore not





visible in Fig. 11 (also not visible in Fig. S6). The lower quartile for '8. Bedidi and Cervelle (1993)' exceed 6.5 wt.% in 11 cases among 24 dust cases (case 3, 4, 5, 6, 7, 13, 18, 19, 21, 22, 23).

Based on the same logic, hematite refractive index of '4. Gillespie and Lindberg (1992)', '5. Hsu and Matijevic (1985)',
'7. Longtin et al. (1988)', '10. Kerker et al. (1979)', and '11. Marusak et al. (1980)' do not produce reasonable results. For example, in dust case 22 and 23 over Australia, the median values of iron-oxide content already exceed 6.5 wt.%. In case 3, 19, 21, the upper quartile of the iron-oxide content is close to 6.5 wt.%.

We exclude '9. Sokolik and Toon (1999)' from the most suitable hematite refractive index. '9. Sokolik and Toon (1999)' cited Querry (1978) for hematite, but Querry (1978) measured complex refractive index of limestone, not hematite. It is
suspected that '9. Sokolik and Toon (1999)' misquoted '6. Querry (1985)' (Zhang et al., 2015), though there are slight differences between '9. Sokolik and Toon (1999)' and '6. Querry (1985)'.

Consequently, our analysis suggests that the remaining hematite refractive index of '1. Chen and Cahan (1981) –1', '2. Chen and Cahan (1981) –2', '6. Querry (1985)', '12. Vernon (1962)', and '13. Scanza et al., (2015)' are the most plausible (summarized in Fig. S7). These results are in agreement with Schuster et al. (2016) selecting Chen and Cahan (1981) as the
baseline refractive index of hematite. Li et al. (2019) paper used both Scanza et al. (2015) and Longtin et al. (1988) as a representative of iron oxides, mentioned the large range of hematite refractive indices, but did not search for the most plausible one.

It is worth to mention that we cannot narrow down the most plausible hematite refractive index with the corresponding cost function only. This is because we fit the exponentially decreasing imaginary refractive index of MAIAC EPIC with
wavelength at 340, 388, 443, and 680 nm derived from equation (10) parametrized only with $b$ and $k_0$. Some of the hematite refractive indices, such as '5. Hsu and Matijevic (1985)', '7. Longtin et al. (1988)', and '10. Kerker et al. (1979)', have exponentially decreasing imaginary refractive index at 350–400 nm, therefore the minimum cost function is always less than 0.003 (in Fig. S10). On the contrary, hematite refractive indices, such as '1. Chen and Cahan (1981) –1', '2. Chen and Cahan (1981) –2', '6. Querry (1985)', '9. Sokolik and Toon (1999)', and '13. Scanza et al. (2015)' (used in this study), have increasing
imaginary refractive index at 350–400 nm, therefore the magnitude of cost function (Fig. S10) is proportional to hematite volume fraction (Fig. S8).

## 5 Conclusions

In our knowledge, this is the first attempt to infer hematite and goethite concentrations, primary components of iron oxides, from the single-viewing satellite measurements. Based on the rationale that hematite and goethite are the major components causing absorption in the UV–Vis region, we used the Maxwell–Garnett effective medium approximation method with the





EPIC spectral aerosol absorption information $k_0$ and $b$. Currently, this algorithm uses pure-dust cases and may be biased in regions/seasons where dust aerosol is mixed with biomass burning smoke. Our retrieval patterns were generally consistent in
time and space with those of previous studies over main global dust source areas.

Sites in the central Sahara and Sahel, such as Libya, Algeria, Mauritania, Niger, Mali, and Bodélé, and in the Middle East such as Saudi Arabia and Kuwait, exhibit greater temporal variability than other sites, implying large dynamic dust-source variability in the Sahara, Sahel, and Middle East. Over the central Sahara and Sahel regions, goethite was generally prevalent except for during the summer monsoon season of May–July, when hematite was predominant and likely sourced from near
Algeria, Mali, and Sudan. The Bodélé Depression area has distinctive lower iron-oxide concentration on climatology map (~1 wt.%) with high AOD throughout the year. Over the Middle East, goethite was prevalent in the Shamal season, whereas hematite was widespread in the atmosphere during March–May. Over India, higher hematite concentrations were detected in 2018 during the pre-monsoon season. The Taklimakan Desert was active except for in winter, with hematite/goethite ratios of ~0.55 (near published values). The Gobi Desert also had a ratio of 0.55 but was active only during spring. Over North America
and Australia, pure-dust-episode analysis indicated hematite-dominated patterns, consistent with earlier studies. Our results, including the 24 representative dust cases over different continental area, clearly implied that the composition of hematite to goethite may differ case by case depending on different source regions, meteorological condition causing transportation, even within a day.

We compared our retrieval data with the Di Biagio et al. (2019) who undertook only soil analyses. While our retrievals
may include both local and long-range-transport contributions of iron oxides; our results cover the range of soil data both qualitatively (relative comparisons site by site) and quantitively. This implies that our retrieval data are robust. There seemed to be better agreement during seasons of local erosion without long-range transport. Further quantitative comparisons are needed with widespread ground-based data such as AERONET, if it becomes available.

We also selected the most plausible hematite refractive indices, given that the maximum value of iron-oxide content that
we have found with *in situ* measurement papers is about 6.5 wt.%. The narrowed-downed hematite refractive indices, such as Chen and Cahan (1981), Querry (1985), Vernon (1962), and Scanza et al. (2015), exhibited a similar range and spectral shape, and are in agreement with Schuster et al. (2016).

The developed algorithm can be applied for other on-orbit instruments with UV–Vis channels. The reported results from EPIC for the first time provide the global over land climatology of hematite and goethite concentrations in mineral dust, and
should be useful for the estimation of SW dust DRE in Earth-system and climate modeling.





**Appendix A**

**Figure A1: Dust episodes over the northern Sahara (rows 1 and 2) and near the Red Sea (rows 3 and 4). From left, RGB TOA image generated from EPIC L1B version 3 data, AOD and SSA at 443 nm, mass concentration of hematite and goethite for each case: Case 4 (row 1), dust episode on February 21, 2016; Case 5 (row 2), dust episode on February 22, 2017; Case 6 (row 3), dust episodes on August 6, 2015; Case 7 (row 4), dust episodes on August 7, 2015.**


**Figure A2: Dust episodes over the Middle East. From left, RGB TOA image generated from EPIC L1B version 3 data, AOD 443 nm, SSA 443 nm, hematite mass concentration, and goethite mass concentration for each case: Case 8 (row 1), dust episode on September 1, 2015; Case 9 (row 2), dust episode on September 2, 2015; Case 10 (row 3), dust episodes on September 7, 2015; Case 11 (row 4) dust episode on July 28, 2018.**

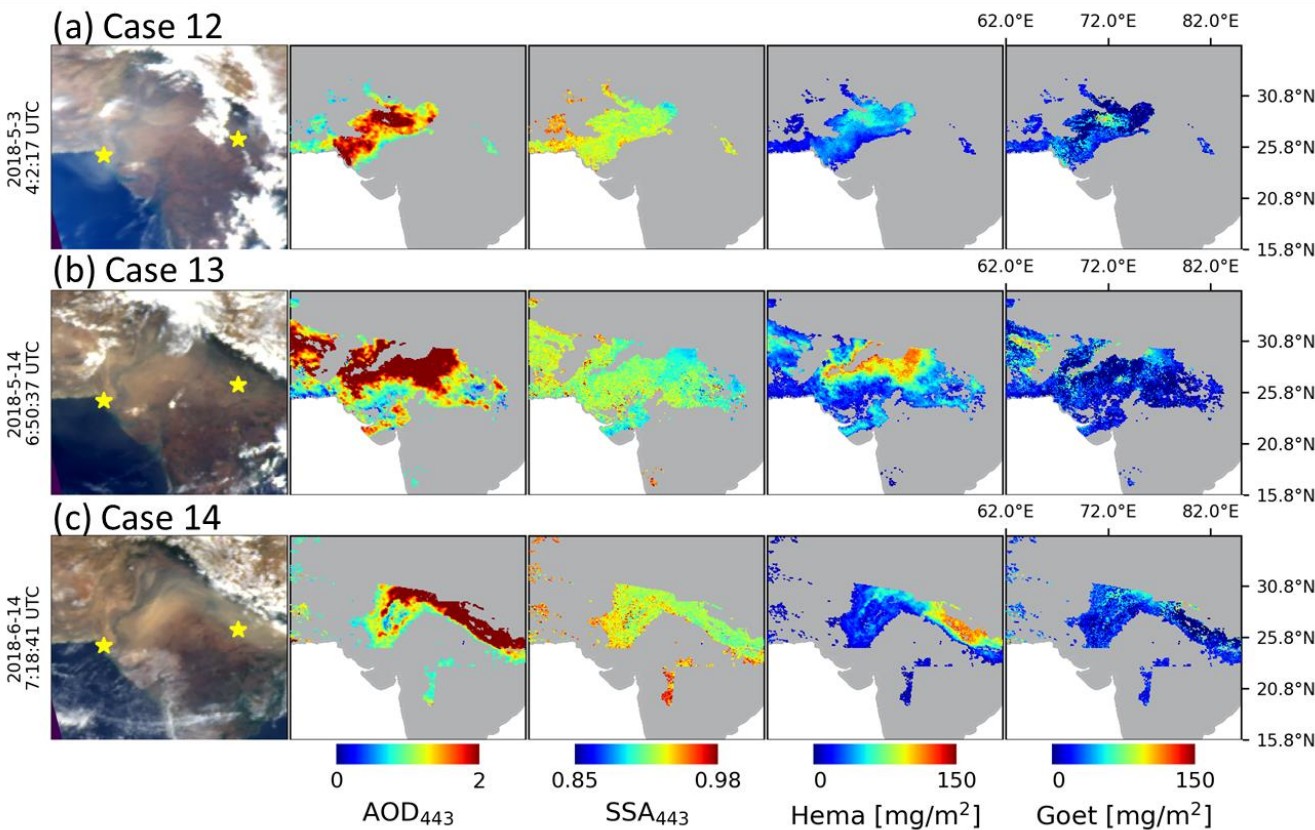

**Figure A3: Dust episodes over India. From left, RGB TOA image generated from EPIC L1B version 3 data, AOD 443 nm, SSA 443 nm, hematite mass concentration, and goethite mass concentration for each case: Case 12 (row 1), dust episode on May 3, 2018; Case 13 (row 2), dust episode on May 14, 2018; Case 14 (row 3), dust episode on June 14, 2018. Yellow stars indicate AERONET sites at Karachi (24.946°N, 67.136°E) and Kanpur (26.513°N, 80.232°E).**



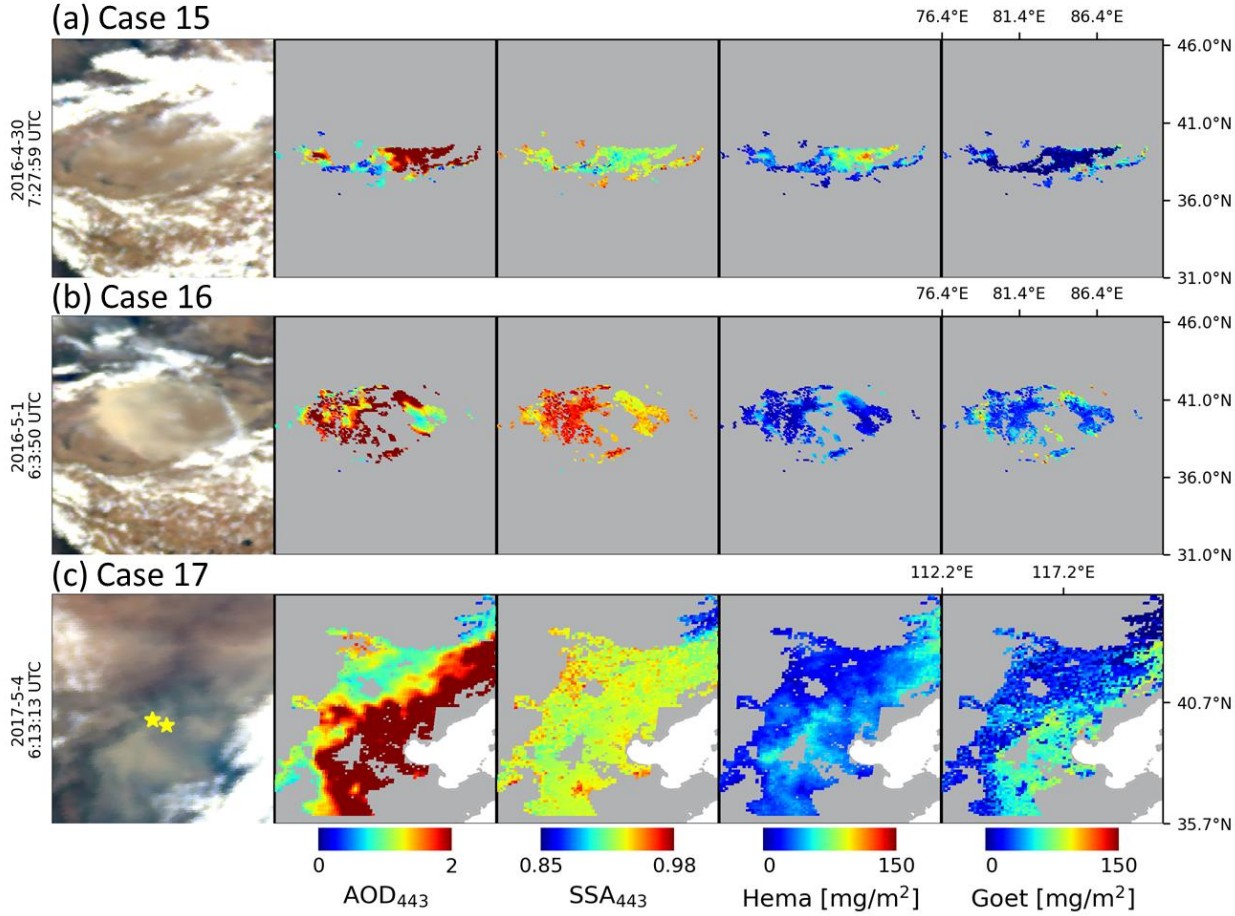

**Figure A4: Dust episodes over the Taklimakan (rows 1 and 2) and Gobi (row 3) deserts. From left, RGB TOA image generated from EPIC L1B version 3 data, AOD 443 nm, SSA 443 nm, hematite mass concentration, and goethite mass concentration for each case: Case 15 (row 1), dust episode on April 30, 2016; Case 16 (row 2), dust episode on May 1, 2016; Case 17 (row 3) dust episode on May 4, 2017. Yellow stars over Case 17 (row 3) indicate AERONET sites at Beijing (39.977°N,116.381°E), Beijing_RADI (40.005°N,116.379°E), and XiangHe (39.754°N,116.962°E).**

**Figure A5: Dust episodes over North America near New Mexico, USA. From left, RGB TOA image generated from EPIC L1B version 3 data, AOD 443 nm, SSA 443 nm, hematite mass concentration, and goethite mass concentration for each case: Case 18 (row 1), dust episode on March 31, 2017, 19:00 UTC; Case 19 (row 2), dust episode on March 31, 2017, 20:00 UTC; Case 20 (row 3), dust episode on April 10, 2019, 20:00 UTC; Case 21(row 4), dust episode on April 10, 2019, 22:00 UTC.**




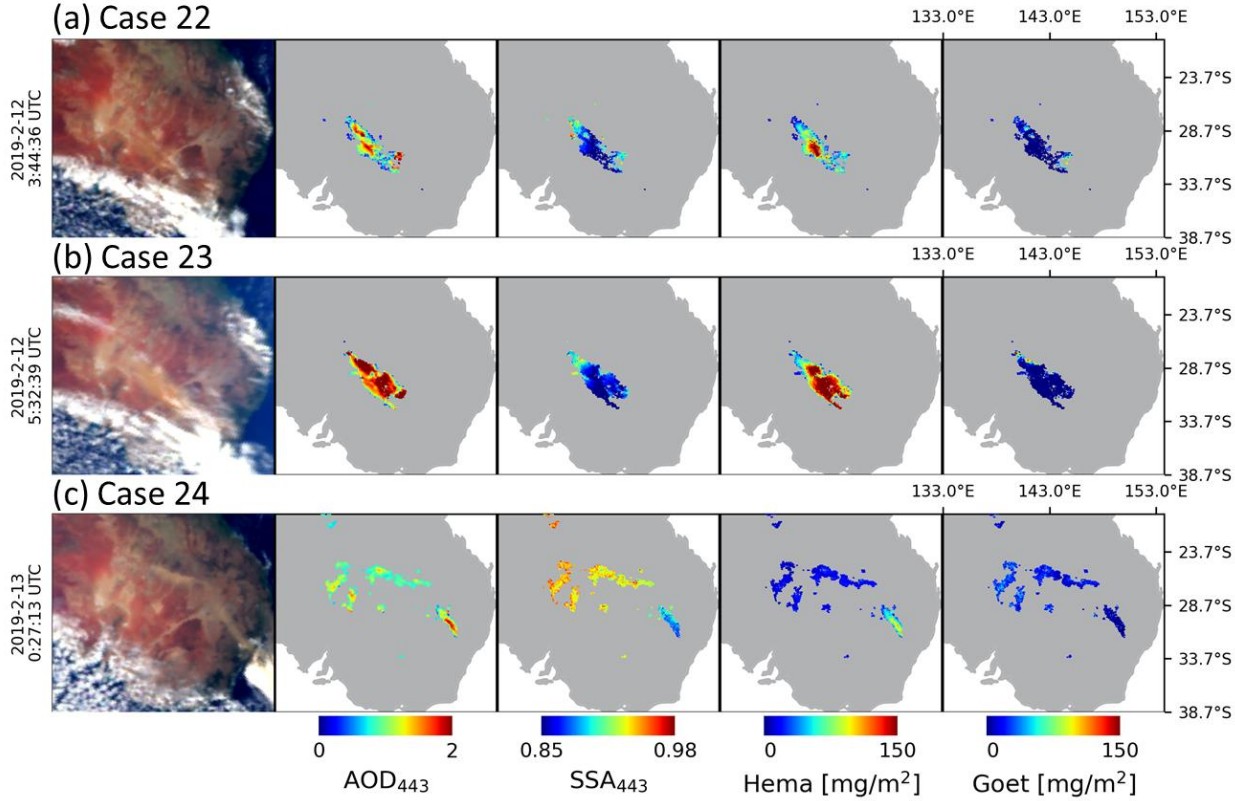

**Figure A6: Dust episodes over Australia. From left, RGB TOA image generated from EPIC L1B version 3 data, AOD 443 nm, SSA 443 nm, hematite mass concentration, and goethite mass concentration for each case: Case 22 (row 1), dust episode on February 12, 2019, 03:00 UTC; Case 23 (row 2), dust episode on February 12, 2019, 05:00 UTC; Case 24 (row 3), dust episodes on February 13, 2019.**






**Author contributions.**

SG and AL designed the study with discussions with GLS and MC. GLS provided major guidance on the algorithm development and data analysis. AL provided the MAIAC EPIC products. OK participated in the collection of hematite refractive indices data. SG, AL, and MC developed the code and performed the retrievals. SG, AL, GLS, and MC analyzed results. SG and AL wrote the manuscript with comments from GLS, MC, PG, MC, OK, OD, JK, ADS, BH, and JSR.

**Competing interests**.

The authors declare that they have no conflict of interest.

**Acknowledgements.**

The work of A. Lyapustin, S. Go, and M. Choi was funded by the NASA DSCOVR program (manager Dr. R. Eckman) and in
part by the NASA PACE program (19-PACESAT19-0039). We are grateful to the AERONET team for providing validation data and to the NASA Center for Climate Simulations providing resources for the EPIC data processing. Co-author J. S Reid was supported by the Office of Naval Research Code 322.





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
