# Peer review of "Inferring iron oxides species content in atmospheric mineral dust from DSCOVR EPIC observations"

_Atmospheric Chemistry and Physics, 2021_

## Author Response (AR1)

**[Reviewer 1]**

**Review comments on "Inferring iron oxides species content in atmospheric mineral dust from DSCOVR EPIC observations" by Go et al. submitted to ACP**

We appreciate the referee's thoughtful reading, valuable suggestions and time that we hope helped us to improve the manuscript. Our point-to-point replies are presented below.

*&&&GENERAL COMMENTS&&&*

The iron oxides contents of goethite and hematite in mineral dust play a key role to quantify the dust light absorption, and then influence its radiative effect. Even though the different spectral behavior of refractive indices, the direct retrieval of goethite and hematite concentration from remote sensing measurements is difficult due to the limited information content. This paper, based on the existing EPIC MAIAC product of aerosol type and spectral imaginary refractive indices, proposed a method to infer columnar goethite and hematite mass/volume concentrations by fitting the EPIC MAIAC spectral (UV-Vis) imaginary refractive indices assuming the Maxwell-Garnett effective medium mixture of non-absorbing host and absorbing hematite and goethite. The results are evaluated with in situ measurements. Overall, this study is well-written and within the scope of ACP. I would recommend this paper to be published in ACP after some comments and concerns being addressed.

1. Since it's a sequential approach relying on the product of EPIC MAIAC spectral (UV-Vis) imaginary refractive indices (k), the quantitative validation of spectral k product is critical while not included in this study. The SSA validation by Lyapustin et al. (2021, FRS) may imply the quality of k. However, I would suggest to perform the validation of EPIC MAIAC k product directly and to quantify the uncertainty of derived hematite and goethite concentrations due to input k uncertainty.

   **Response:** We agree that validation of the spectral imaginary refractive index is important. However, any quantitative comparison of $k(\lambda)$ with AERONET, as mentioned in Lyapustin et al. (2021), is associated with high uncertainty. First, AERONET states a 30-50% accuracy for the imaginary refractive index at AOD440 nm > 0.4 (Dubovik et al., 2000), with uncertainties being higher for the coarse mode dust as well as for optically thin aerosols (lower AOD). Second, the main sensitivity of MAIAC for SAE (parameter b) comes from the 340-443 nm range, whereas AERONET provides spectral dependence of refractive index for the non-overlapping range of wavelengths 440-1020 nm. For these reasons, Lyapustin et al., (2021, Lines 156-161) provided a qualitative evaluation of spectral imaginary refractive index $k(\lambda)$.

   Nevertheless, to address the Reviewer's question, we conducted a direct comparison of parameters $k_0$ and b with AERONET. The AERONET "parameter b" was derived from $k(\lambda)$ values at 440nm and 680nm. The results are shown in the figure below for the (R1) northern Africa, (R2) the Sahel, (R3) East Africa and the Middle East, (R4) central Asia, (R5) East Asia. In case of $k_0$, 95.7% (R1), 100% (R2), 100% (R3), 43.8% (R4), 73.3% (R5) of the MAIAC EPIC $k_0$-values are within the expected error (±0.003). Regarding b values, 32.9% (R1), 25% (R2), 0% (R3), 61% (R4), 63.3% (R5) of the MAIAC EPIC b-values are within the expected error (±0.5). A pixel-level assessment of the uncertainty for the derived hematite and goethite concentrations is currently under development for the next version of the MAIAC EPIC algorithm.

[Figure]

Fig. 1. Figure R1. Validation results of AOD at 443 nm, SSA at 443 nm, k at 680 nm, b values of MAIAC EPIC with AERONET data for the (first row) total, (R1; second row) northern Africa, (R2; third row) the Sahel, (R3; fourth row) East Africa and the Middle East, (R4; fifth row) central Asia, (R5; sixth row) East Asia. Spatial and temporal collocation criteria within 30 km and 3h are used.

2. In my view, EPIC MAIAC dust detection is another key information used to select pixels and perform the hematite and goethite inversion. The authors claimed that some dust + mixed/smoke aerosol cases, which are classified into dust, may affect the retrievals for some specific regions. In this sense, I would encourage to include the maps of AE in the analysis which may provide some insights for dust detection.
   - Response: Thank you for the comments. Because EPIC is a single-view instrument with insufficient information content, MAIAC does not retrieve the Ångström Exponent (AE) over land. The dust retrievals are performed for (AOD, $k_0$, b) assuming real refractive index, spheroidal model and fixed size distribution.

3. How do you evaluate the derived columnar goethite and hematite mass concentration with *in-situ* near-surface measurements? How do you assume the vertical distribution? Please clarify in the text.
   - Response: We assume the vertical distribution with a 2km box shaped profile in MAIAC look-up tables. For instance, the boundary-layer aerosol is assumed uniformly distributed in 0-2km layer for 1km effective height. In case of 4km, the 90% of aerosol is assumed at 3-5km, with 10% below. We added the following sentence (line 146):
   "MAIAC retrievals are reported for the effective aerosol heights of 1 km and 4 km with 2km box-shaped profile, representing the typical boundary layer and free troposphere transported aerosol.".
   Since the retrieval results are columnar goethite and hematite mass concentration, we did not compare the goethite and hematite mass concentrations directly with the in-situ near-surface measurements. Instead, iron oxides mass fraction to the column-integrated total mass concentration was compared with in-situ measurements as in Fig 8. Di Biagio et al. (2019). Di Biagio et al. (2019) analyzed aerosols generated from natural soil samples.

4. The quality/clarity of the figures (e.g., Figures 1, 5, A1, S1) can be improved.
   - Response: Corrected. Regarding Figure 5, A1, the resolution of the RGB image cannot be improved.

**&&&SPECIFIC COMMENTS&&&**

- Figure 1: the quality/clarity of the figure should be improved.
  Response: Corrected. Regarding Figure 5, A1, the resolution of the RGB cannot be improved.

- Line 107: a recent study by Wang et al. (2021, 10.1016/j.atmosenv.2020.117959) developed an algorithm to derive aerosol components from effective density and spectral refractive indices.

  Response: Thank you for the information. The reference is added (line 110).

- Section 2.1: You mentioned the validation results of MAIAC EPIC SSA. Have you evaluated the MAIAC EPIC imaginary part of refractive index (k), since the fitting of MAIAC EPIC k is then used to derive hematite and goethite?

Response: This question was addressed above (see General comments, #1).

- Line 180: the link does not work, please check.

  Response: Yes, we are aware of that, but this is the exact link that Scanza et al. (2015) provided.

- Line 215: why don't you fit real part together with imaginary part?

  Response: MAIAC EPIC assumes the real part of the refractive index in the (AOD, $k_0$, b) retrievals, it does not retrieve $n_{rtr}$. However, as mentioned in line 214-216, the imaginary refractive index of mixture ($k_{mix}$) is a function of both real and imaginary refractive index of the inclusions 1, 2, and host. Therefore, realistic values of the real refractive indices of inclusions 1, 2, and the host are still required along with the imaginary part. This assumption will be included as part of our uncertainty analysis in the future.

- Line 227: what do you mean flexible retrievals?

  Response: Flexible retrieval means the retrieval algorithm of {AOD, $k_0$, b}. To improve clarity of presentation, we deleted the word "flexible" and added additional reference Lyapustin et al. (2021).

- Equations 11-13: So, you use the coarse mode volume concentration to approximate the total, right? Then the $C_{v, hema} = C_{vc} \times f_{hema}$ is named as the hematite total volume concentration. Is the coarse mode hematite volume concentration more precisely?

  Response: That's true. We changed the "total" to the "coarse mode".

- Line 283: Could you explain why the algorithm does not retrieve dust aerosol over South America and southern Africa?

  Response: With EPIC lacking bands beyond 780nm, MAIAC cannot differentiate between the smoke and dust despite it detects absorbing aerosols. As smoke is much more ubiquitous aerosol type, MAIAC makes dust retrievals only over pre-defined dust regions. At present, South America and southern Africa are not designated as dust regions in MAIAC EPIC.

- Figure 5: There are more cloud-free pixels in the RGB images than that in the AOD, SSA, Hema and Goet plots. Could you explain how these pixels are selected?

  Response: For reliable retrievals, we only select pixels with AOD>0.6 (Ln. 230). At lower AOD, the retrieval error grows rapidly.

- Line 449: How do you convert hematite and goethite volume / mass concentrations to the iron-oxide mass fraction? sum of them? It seems missed in the methodology.

  Response: We calculated $C_{M,hem}$ , $C_{M,goet}$, $C_{M,host}$ values through equation (14). Then the iron-oxide mass fraction can be calculated with $(C_{M,hem} + C_{M,goet}) * 100 / (C_{M,hema} + C_{M,goet} + C_{M,host})$. I added the following sentences at line 464.

  "Iron-oxide mass fraction is calculated as $(C_{M,hema} + C_{M,goet}) \times 100 / (C_{M,hema} + C_{M,goet} + C_{M,host})$."

- Line 469: with a size range of 2-6 days? Please check.

Response: The sentence is reworded to "a time span typical of 2 to 6 days of transport" at line 475.

- Line 486: The colocation method should include also in the main text. What's the spatial resolution of the product? Why the spatial window is different over Australia (+/-3 degree)?

  Response: The spatial resolution of the MAIAC EPIC product is 10 km (~0.1 degree) (Ln. 145). To build the monthly variation plot for hematite and goethite (Figure 8), we used the spatial window (± 1°) (Ln. 492). Over Australia, the window was expanded to ±3° in order to accumulate a sufficient number of retrievals. We added the following sentence to describe this (Ln. 477-478):

  "Each site represents the area of ± 1° of MAIAC EPIC data except for Australia where the box size was expanded to ± 3° to accumulate enough retrievals."

- Line 586: Why do you use the threshold AOD>1.0? Is it for dust detection?

  Response: Satellite-data pixels with $AOD_{443} > 1.0$ were used in the average, as some pixels of goethite retrieval with $AOD_{443} < 1.0$ display "noise" or "blob" patterns (for example, see third and fourth rows, Fig. A1). This empirical threshold typically guarantees lack of noise. To explain this, we added the following sentence (Ln. 595):

  "As in Fig. 8, pixels of AOD > 1.0 were used to compute the average, as some pixels of goethite retrieval with AOD < 1.0 display "noise" or "blob" patterns."

- Line 589: Since it may contain some mixed / smoke aerosol cases, I would suggest to include maps of AE in Fig S2-S5 that may provide some insights for aerosol types.
  Response: Please, see our response to question 2 of General Comments.

- Line 634: real refractive indices -> real part of refractive indices

  Response: Corrected.

- Line 708: the reliable product of spectral imaginary part of refractive indices is a precondition.
  Response: Please, see our response to question 1 of General Comments.

**[References]**

Di Biagio, C., Formenti, P., Balkanski, Y., Caponi, L., Cazaunau, M., Pangui, E., Journet, E., Nowak, S., Andreae, M. O., Kandler, K., Saeed, T., Piketh, S., Seibert, D., Williams, E., and Doussin, J.-F.: Complex refractive indices and single-scattering albedo of global dust aerosols in the shortwave spectrum and relationship to size and iron content, Atmos. Chem. Phys., 19, 15503–15531, https://doi.org/10.5194/acp-19-15503-2019, 2019.

Dubovik, O., Smirnov, A., Holben, B. N., King, M. D., Kaufman, Y. J., Eck, T. F., & Slutsker, I. (2000). Accuracy assessments of aerosol optical properties retrieved from Aerosol Robotic Network (AERONET) Sun and sky radiance measurements. *Journal of Geophysical Research: Atmospheres*, *105*(D8), 9791-9806.

Lyapustin, A., Go, S., Korkin, S., Wang, Y., Torres, O., Jethva, H., & Marshak, A. (2021). Retrievals of Aerosol Optical Depth and Spectral Absorption from DSCOVR EPIC. *Frontiers in Remote Sensing*, *2*, 7.

Scanza, R. A., Mahowald, N., Ghan, S., Zender, C. S., Kok, J. F., Liu, X., ... & Albani, S. (2015). Modeling dust as component minerals in the Community Atmosphere Model: development of framework and impact on radiative forcing. *Atmospheric Chemistry and Physics*, *15*(1), 537-561.

**[Reviewer 2]**

This is a very valuable study that describes a methodology of deriving hematite and goethite concentrations from the UV-VIS single-viewing space imager EPIC aboard the DSCOVER. Obviously, deriving such detailed aerosol property as hematite and goethite concentrations from this type of space sensor requires numerous hypothesis and a priori information, however, the authors do quite a careful work on their educated definition. The methodology is then applied for a series of carefully selected case studies that fulfill the conditions for an optimal application of the methodology. The study also presents a revision of a large diversity of the refractive indexes of hematite and goethite reported in literature, and an analysis justifying the choice made in the study. Elements of validation of the obtained hematite and goethite concentrations using *in situ* data are presented as well, which is not an evident task for satellite derived aerosol property. I would like also to acknowledge the work done on revision of the literature. The study is certainly in the scope of ACP, it is a solid work and indeed worth the publishing. I have just a few main questions which I believe addressing could strengthen the study, but I leave to the authors to decide if to include the elements of reply in the manuscript or not.

The evaluation of the uncertainty in the derived percentage of iron oxide due to different refractive indexes assumed is very interesting and informative (Fig. 11). I would think about at least two other assumptions that require evaluation of the uncertainty in the derived concentrations.

- First, the real part of the refractive index, which is fixed. However, generally, the values of real and imaginary parts are related and the imaginary part is varying here. Specially, the real part of iron oxides is quite high (Fig. 1, a), so the real part of mixture is expected to vary quite a bit depending on the iron oxide fraction. I would suggest calculating the effective refractive index and simulate satellite signal for internal mixture of nonadsorbing dust host and iron oxides for the corresponding real and imaginary refractive indexes and then deriving (under the fixed real part assumption) the hematite and the goethite concentrations for this synthetic signal. How will it compare to the initially used hematite and goethite fractions? Indeed, it is mentioned in the paper that Di Biagio et al. 2019 conclude that the real part is generally source- and wavelength-independent with a range of 1.48-1.55, but this range seems to be big enough to cause the derived iron oxides fractions variability.

➔ Thank you for the valuable suggestion. The below figure (Figure R1-R3) shows the refractive index of mixture (based on the Maxwell-Garnett internal mixing rule) with respect to the iron oxide fraction which shows the sensitivity indirectly. As you mention, the real part of the mixture is expected to vary quite a bit depending on the iron oxide fraction. At 443 nm, the real part of the mixture varies with a range of 1.51-1.59, and the imaginary part of the mixture varies with a range of 0.0-0.03 for certain volume fractions of hematite (0–0.05) and goethite (0–0.03). The current MAIAC algorithm does not retrieve real part of the refractive index due to insufficient information contents, and the related sensitivity study regarding the real part of the refractive index is beyond the scope of this study. We will evaluate this uncertainty in the follow-on analysis.

[Figure]

Fig. 1. Figure R1. Theoretical retrieval test. The real part of refractive index of mixture at 443 nm as a function of the hematite volume fraction (x axis) and goethite volume fraction (y axis) of MAIAC EPIC parameter using 13 models of hematite refractive index listed in Table 1.

[Figure]

Fig. 2. Figure R2. Theoretical retrieval test. The imaginary part of refractive index of mixture at 443 nm as a function of the hematite volume fraction (x axis) and goethite volume fraction (y axis) of MAIAC EPIC parameter using 13 models of hematite refractive index listed in Table 1.

[Figure]

Fig. 3. Figure R3. Theoretical retrieval test. The real part (top row) and imaginary part (bottom row) of refractive index of mixture at 340, 388, 443, 680 nm as a function of the hematite volume fraction (x axis) and goethite volume fraction (y axis) of MAIAC EPIC parameter.

- Second, 1 km aerosol height is assumed. The assumption is justified by generally good agreement of EPIC derived absorption with AERONET, but this is on average and fluctuations in specific cases are expected. The dust over hot desert surfaces, specially over Sahara, is lifted to rather higher than 1 km altitudes and sensitivity of UV to the dust altitude is known. What if to conduct a similar test as in the case of real refractive index? That is, to simulate the satellite signal for dust at different altitudes and invert for the iron oxides fractions using the fixed (1 km) dust altitude. These exercises can evaluate uncertainty in the derived fractions due to these two assumptions and provide a valuable error bar. The effect of presence of carbonaceous aerosols (mixture with smoke) can be evaluated in the same manner.

Response: Thank you for the important comments. We are planning to provide the hematite/goethite contents for both 1km and 4km in the next version of the MAIAC EPIC. The effect of the presence of carbonaceous aerosols (mixture with smoke) will be also evaluated as a future study. The figure below shows the effect of assumed dust layer height (1km, 4km) on derived iron oxide fractions for the dust case 8. The iron oxide fraction tends to decrease for the elevated dust layer due to decreased b and $k_0$ values. In other words, for elevated dust aerosol it takes less absorption to create the same reduction of the top-of-atmosphere reflectance in the UV.

We can mention two points in this regard:

1) As EPIC also has O2 A- and B-bands sensitive to aerosol profile, we initiated the work to expand MAIAC EPIC algorithm and provide a suite of (AOD, $k_0$, b, and effective height) from the UV-Vis-NIR measurements including A- and B-bands;

2) We are also developing a framework to evaluate pixel-level retrieval uncertainty which will be provided in the future. Currently, such analysis is beyond the scope of this paper.

[Figure]

Fig. 4. Figure R4. Box and whisker plot of iron-oxide content by mass (wt.%; y axis) for 13 models of hematite refractive index for the dust cases 8 (left: 1 km, right: 4 km). Red horizontal dashed line on each figure shows the maximum expected iron-oxide content (6.5 wt.%) based on in situ measurements.

**[Reviewer 3]**

This work presents the application and discuss the results of a new algorithm that applies to MAIAC EPIC data to retrieve the mass concentration of hematite and goethite content in the global atmosphere over land. The aim of the study is of relevance for dust investigations, from the earth's radiative budget analyses to the biogeochemical studies. The retrieval scheme and the Maxwell Garnett approximation applied seem reasonable. Hypothesis and steps of the retrieval are clearly presented. Data from laboratory and field observations are used to compare and validate the results. Regional and seasonal variability of hematite and goethite mass concentration retrievals are analyzed. The paper i well written and properly organized.

I am in favour of the publication of the paper after the authors have addressed some points below.

- The validation of k from MAIAC, which is the starting point of the analysis, is actually missing. Can you compare the k values against lab and field observations as you did for the retrieved hematite and goethite mass concentrations?
  Response: We agree that validation of the spectral imaginary refractive index is important. However, any quantitative comparison of $k(\lambda)$ with AERONET, as mentioned in Lyapustin et al. (2021), is associated with high uncertainty. First, AERONET states a 30-50% accuracy for the imaginary refractive index at $AOD440$ nm > 0.4 (Dubovik et al., 2000), with uncertainties being higher for the coarse mode dust as well as for optically thin aerosols (lower AOD). Second, the main sensitivity of MAIAC for SAE (parameter b) comes from the 340-443 nm range, whereas AERONET provides spectral dependence of refractive index for the non-overlapping range of wavelengths 440-1020 nm. For these reasons, Lyapustin et al., (2021, Lines 156-161) provided a qualitative evaluation of spectral imaginary refractive index $k(\lambda)$.
  Nevertheless, to address the Reviewer's question, we conducted a direct comparison of parameters $k_0$ and b with AERONET. The AERONET "parameter b" was derived from $k(\lambda)$ values at 440nm and 680nm. The results are shown in the figure below for the (R1) northern Africa, (R2) the Sahel, (R3) East Africa and the Middle East, (R4) central Asia, (R5) East Asia. In case of $k_0$, 95.7% (R1), 100% (R2), 100% (R3), 43.8% (R4), 73.3% (R5) of the MAIAC EPIC $k_0$-values are within the expected error (±0.003). Regarding b values, 32.9% (R1), 25% (R2), 0% (R3), 61% (R4), 63.3% (R5) of the MAIAC EPIC b-values are within the expected error (±0.5). A pixel-level assessment of the uncertainty for the derived hematite and goethite concentrations is currently under development for the next version of the MAIAC EPIC algorithm.

[Figure]

Fig. 1. Figure R1. Validation results of AOD at 443 nm, SSA at 443 nm, k at 680 nm, b values of MAIAC EPIC with AERONET data for the (first row) total, (R1; second row) northern Africa, (R2; third row) the Sahel, (R3; fourth row) East Africa and the Middle East, (R4; fifth row) central Asia, (R5; sixth row) East Asia. Spatial and temporal collocation criteria within 20 km and 3h are used.

- To my understanding the algorithms applies only on cases with AOD larger than 0.6, when MAIAC EPIC provides outputs of k, AOD and b. This is one point to discuss when analyzing the regional/seasonal concentrations of hematite and goethite, perhaps. What is the impact of this assumption on the retrieval and its exploitation/application? Is it a real "global" climatology that is obtained?

  Response: Thank you for this comment, it is indeed not sufficiently explained in the text. Yes, currently the algorithm only applies the retrieval for pixels of AOD>0.6. And, for the global climatology (Section 3.3), pixels of AOD>1.0 were used to compute the average due to the low sensitivity when pixels of AOD<1.0. This could create some differences due to the omission of the fine mode dust, which contains hematite and goethite in the fine mode (clay) according to some publications (e.g., Journet et al., 2014; Menut et al., 2020).

  Below is the figure from Journet et al. (2014) that illustrates the (b) hematite in the clay (diameters Dp=0-2 μm) fraction, (c) goethite in the clay fraction, and (d) goethite in the silt (diameters Dp=2-50 μm) fraction.

  The hematite content in the clay fraction is usually below 1.5% but reaches 5% in some regions, including the longitudinal band from Montana to Texas in the US, a latitudinal band across southern Russia, and arid regions of northern Africa, while soils in southern Brazil/northern Argentina have a high hematite content exceeding 5%. Goethite occurs in both the clay- and silt-sized fractions. The amount of goethite in the clay fraction is generally higher than the amount of hematite, and more variable (from 0 to 15%). Goethite is generally more abundant in humid tropical environments while hematite becomes more abundant in the seasonally dry tropics.

  I added the following sentences at Line 622-628.

  "The seasonal average EPIC data is based on the AOD larger than 1.0, and this may cause the omission of fine mode dust such as clay fraction of hematite (Journet et al., 2014; Menut et al., 2020). Journet et al. (2014) mentioned that the hematite content in the clay fraction is usually below 1.5% but reaches 5% in some regions, including the longitudinal band from Montana to Texas in the US, a latitudinal band across southern Russia, and arid regions of northern Africa, while soils in southern Brazil/northern Argentina have a high hematite content exceeding 5%. Error or uncertainty associated with the omission of the fine mode dust is beyond the scope of this study, and will be provided in the next version of the MAIAC EPIC algorithm."

[Figure]

**Fig. 7.** Iron and iron oxides in soils: **(a)** iron in the clay fraction (CASE 1), **(b)** hematite in the clay fraction (CASE 1), **(c)** goethite in the clay fraction (CASE 1) and **(d)** goethite in the silt fraction.

Fig. 2. Figure R2. Iron and iron oxides in soils: (a) iron in the clay fraction (CASE 1), (b) hematite in the clay fraction (CASE 1), (c) goethite in the clay fraction (CASE 1) and (d) goethite in the silt fraction (Journet et al., 2014).

- This retrieval applies to the AOD for the coarse fraction, therefore hematite and goethite are referring to the coarse dust AOD only. Is this assumption biasing the retrieval in such a way? Considerations on the size-dependent composition of dust and distribution of Hematite and Goethite as a function of size should be added.

Response: Thank you for the valuable comments. The size distribution we assumed in the retrievals will always be an issue unless we have more information contents (e.g., multi-angle measurements, polarization). There is always a small fraction of the fine mode, but it is irrelevant for the dust storms at high AOD. The dust size distribution model we are using is from Oleg Dubovik's 2003 JAS paper (based on AERONET data at Solar Village). Based on AOD validation (in MAIAC MODIS and EPIC), it works well. The model has a bi-lognormal size distribution with Rv_f=0.12, Sigv_f=0.5, Rv_c=1.9, Sigv_c=0.6 for the fine and coarse modes. The volumetric concentration for the coarse mode increases with AOD, and is constant for the fine mode: Cv_f=0.02, and Cv_c=0.02+0.9*AOD. Even if we admit more fine mode in the retrievals,

the absorption in the Blue-UV range, and its spectral dependence, is still defined by the dominant coarse mode at the range of AOD>0.6 we are working with. In summary, we do not believe we have a sensitivity to the fine mode with EPIC in case of dust, at least within the size distribution model we are using presently.

- Figure 3 can probably show goethite dataset for real and imaginary refractive indices as well, similar for Table 1, the corresponding information for goethite is missing

  Response: Thank you for the comments. Unlike hematite, information on the complex refractive index of goethite is very scarce. We only found two reports of complex refractive indices of goethite having been published by Bedidi and Cervelle (1993) and Glotch and Roman (2009) for 0.45-0.75 and 8–50 µm wavelength ranges, respectively. Glotch and Roman (2009) wavelength ranges 8-50 µm is out of interest. The refractive index of Bedidi and Cervelle (1993) that we use is described in Figure 1.

- Line 284: what do you mean with « significant » events? in AOD or other?

  Response: Yes, this is significant dust AOD event. Changed to "significant" → "significant AOD".

- Line 291: Di Biagio et al. (2019) analyzed aerosols generated from natural soil samples and not soil properties, check elsewhere in the paper (in example sect 3.2 and following text) and clarify this aspect which is of relevance for the validation of the retrieved atmospheric aerosol hem and goet concentration from the present study.

  Response: Thank you for the comments. I added "Di Biagio et al. (2019) analyzed aerosols generated from natural soil samples" (line 472). The following lines were also corrected accordingly: 127, 219, 294, 473, 475, 512, 527, 562, 567, 587, 717.

- What is the impact of fixing the real refractive index on the retrieval?
- ➔ Response: The figure below shows the refractive index of mixture (based on the Maxwell-Garnett internal mixing rule) with respect to the iron oxide fraction – in a way it shows the sensitivity indirectly. The real part of the mixture is expected to vary quite a bit depending on the iron oxide fraction. The current MAIAC algorithm does not retrieve real part of the refractive index due to the lack of information contents of the single view satellite, and the related sensitivity study regarding the real part of the refractive index is the beyond scope of this study. We will evaluate the impact of this assumption on uncertainty in a future study.

[Figure]

Fig. 3. Figure R3. Theoretical retrieval test. The real part (top row) and imaginary part (bottom row) of refractive index of mixture at 340, 388, 443, 680 nm as a function of the hematite volume fraction (x axis) and goethite volume fraction (y axis) of MAIAC EPIC parameter.

- I would discuss possible perspective applications over ocean, an aspect which could be of great interest for biogeochemical studies

  Response: Thank you for the suggestion. We added the following sentence in the conclusion section:

  Line 727: "In near future, the algorithm will be expanded over the global ocean to support ocean biogeochemical studies (e.g., Tagliabue et al., 2017)."

- Please, clearly state in the abstract the conditions to which this retrieval applies (land, AOD>0.6, ..)

  Response: Corrected.

**[References]**

Bedidi, A., & Cervelle, B. (1993). Light scattering by spherical particles with hematite and goethite like optical properties: effect of water impregnation. *Journal of Geophysical Research: Solid Earth*, *98*(B7), 11941-11952.

Dubovik, O., Smirnov, A., Holben, B. N., King, M. D., Kaufman, Y. J., Eck, T. F., & Slutsker, I. (2000). Accuracy assessments of aerosol optical properties retrieved from Aerosol Robotic Network (AERONET) Sun and sky radiance measurements. *Journal of Geophysical Research: Atmospheres*, *105*(D8), 9791-9806.

Glotch, T. D., & Rossman, G. R. (2009). Mid-infrared reflectance spectra and optical constants of six iron oxide/oxyhydroxide phases. *Icarus*, *204*(2), 663-671.

Journet, E., Balkanski, Y., & Harrison, S. P. (2014). A new data set of soil mineralogy for dust-cycle modeling. *Atmospheric Chemistry and Physics*, *14*(8), 3801-3816.

Lyapustin, A., Go, S., Korkin, S., Wang, Y., Torres, O., Jethva, H., & Marshak, A. (2021). Retrievals of Aerosol Optical Depth and Spectral Absorption from DSCOVR EPIC. *Frontiers in Remote Sensing*, *2*, 7.

Menut, L., Siour, G., Bessagnet, B., Couvidat, F., Journet, E., Balkanski, Y., & Desboeufs, K. (2020). Modelling the mineralogical composition and solubility of mineral dust in the Mediterranean area with CHIMERE 2017r4. *Geoscientific Model Development*, *13*(4), 2051-2071.

Tagliabue, A., Bowie, A. R., Boyd, P. W., Buck, K. N., Johnson, K. S., & Saito, M. A. (2017): The integral role of iron in ocean biogeochemistry. Nature, 543(7643), 51-59.

---

## Author Response (AR2)

**Comments to the author**:
The comments from the three reviewers were well addressed and several Figures from the responses show that you did the extra work to answer their concerns.
To publish this paper into ACP, I simply ask you to make two slight addition/modification to the manuscript.

We appreciate the editor's thoughtful suggestion and valuable time. I corrected the manuscript as you mentioned below.

1- The first one is to include in your paper the answer you made to Reviewer 1 concerning the following question that indicates areas where the EPIC has limitations to retrieve dust:
"Line 283: Could you explain why the algorithm does not retrieve dust aerosol over South America and southern Africa?

Response: With EPIC lacking bands beyond 780nm, MAIAC cannot differentiate between the smoke and dust despite it detects absorbing aerosols. As smoke is much more ubiquitous aerosol type, MAIAC makes dust retrievals only over pre-defined dust regions. At present, South America and southern Africa are not designated as dust regions in MAIAC EPIC."

Response: The response is included at line 285.

2- The second one is on Line 219 of the latest version of the manuscript:
Change
"...based on a study of 19 mineral dust soil-derived aerosol samples from different main global dust source regions."

To
"based on a study of 19 mineral dust aerosol generated from soil samples coming from the main global dust source regions."

Response: Corrected at line 219.